# Multimodal Masked Point Distillation for 3D Representation Learning

**Muhammad Abdullah Jamal**
*Intuitive Surgical Inc.*

**Omid Mohareri**
*Intuitive Surgical Inc.*

**Reviewed on OpenReview:** *https://openreview.net/forum?id=Gxb3z4VlM7*

## Abstract

We propose a two-stage pre-training approach using point clouds for a diverse set of 3D understanding tasks. In the first stage, we pre-train the 3D encoder to acquire knowledge from the other modalities such as vision and language. This stage aligns 3D representations with multiple modalities by leveraging several pre-trained foundation models, unlike the current cross-modal paradigm that typically uses only a single pre-trained model. In the second stage, the pre-training approach is improved upon masked point modeling by global-local feature distillation of semantic 3D embeddings and token shuffling approach. These techniques enable the model to focus on the 3D modality while leveraging the multimodal information associated with the point clouds. This pre-training approach is model-agnostic and can be applied to any 3D transformer encoder. We conduct extensive experiments on a wide range of 3D understanding tasks, from synthetic and real-world object recognition to indoor semantic segmentation and object detection, achieving state-of-the-art results. For instance, on the ScanObjectNN variants, our approach achieves **96.1%**, **94.2%** and **91.2%** accuracy using multi-scale 3D encoder proposed in Point-M2AE.

## 1 Introduction

Self-supervised learning (SSL) approaches have paved the way for the creation of foundation models (Bommasani et al., 2022) that leverage abundant unlabeled data, enabling adaptation to various downstream tasks with only a small amount of labeled data. This paradigm has led to remarkable success across multiple domains, including NLP (Radford & Narasimhan, 2018; Devlin et al., 2019; Wei et al., 2023; Ouyang et al., 2022), 2D vision (He et al., 2022; 2020; Chen et al., 2020), and vision-language (Radford et al., 2021; Alayrac et al., 2022; Li et al., 2023). Nevertheless, the success of these models fundamentally depends on the availability of large-scale pre-training data.

Recently, many successful pre-training strategies have been adapted for 3D point cloud understanding. However, compared to NLP and vision, the 3D domain suffers from a lack of data at scale, as collecting and annotating point cloud data is challenging. This phenomenon, often referred to as a *data desert* (Dong et al., 2022), can hinder the scaling and development of 3D foundation models. Contrastive learning based approaches (Afham et al., 2022; Xie et al., 2020; Sanghi, 2020) that learn instance-discriminative representations by maximizing the similarity of different views of the point cloud, fail to generalize when there is a lack of pre-training data (Qi et al., 2023). On the other hand, generative pre-training (Pang et al., 2022; Zhang et al., 2022a; Zha et al., 2023a) which follows the masked autoencoding (MAE) in 2D, can bring significant improvement with less amount of data. However, the representations learned by these approaches may not fully capture the holistic representations due to the intricate geometric and spatial relationships in 3D data. Since point clouds are unstructured and unordered set of coordinates, there is no straightforward or exact supervision method like one-to-one mean squared error (MSE) loss between the ground truth and the

reconstructed one. MAE-based approaches typically use Chamfer distance as a pre-training loss to compute an approximate matching between two sets of points. This allows for an initial alignment of point clouds, though it still has limitations in capturing fine-grained details or handling unordered nature of point clouds effectively. Additionally, there are several shortcomings to using Chamfer distance as a loss function for point cloud completion (Wu et al., 2021; Liu et al., 2019; Huang et al., 2023a).

Another line of work is cross-modal learning (Xue et al., 2023; Dong et al., 2022; Zhang et al., 2022b; Zhu et al., 2022; Xu et al., 2022; Tang et al., 2024; Qian et al., 2024) which leverages other modalities like image, depth maps and text associated with point cloud data, and has shown impressive performance. In this context, most approaches utilize pre-trained foundation models to align multimodal features with point cloud features, resulting in enhanced 3D representations complemented by multimodal capabilities. However, these methods primarily rely on a single vision or language foundation model, which can limit the feature representation of the multimodal data. CLIP (Radford et al., 2021), which has been mostly used in cross-modal learning, has demonstrated remarkable performance in a zero-shot setting, but it under-performs compared to vision transformer (Dosovitskiy et al., 2021) as reported by ReCon (Qi et al., 2023). CLIP focuses more on global representations, which may not be optimal for dense tasks such as semantic segmentation, as noted by (Li et al., 2022). We hypothesize that by leveraging unique properties from multiple foundation models can further enhance the multimodal representations. For example, compared to CLIP, DINOv2 (Oquab et al., 2023) has emerged with strong spatial features for dense tasks. Segment Anything (SAM) (Kirillov et al., 2023) has shown remarkable performance on segmentation due to dense feature representations. Furthermore, existing cross-modal methods (Huang et al., 2023b; Qi et al., 2023; Xue et al., 2023) rely on limited prompt templates based on category names, which both constrains scalability and reduces the diversity of text descriptions available for pre-training.

To address these issues, we propose a two-stage pre-training approach that can learn holistic 3D representations using rich and well-aligned multimodal data. In the first stage, we pre-train a 3D encoder using multimodal data by leveraging multiple vision and language foundation models using contrastive objective. We follow ULIP-2 (Xue et al., 2024) to generate diverse language descriptions of point cloud data using BLIP-2 (Li et al., 2023), which encapsulates all expressible information about 3D data. We also render 2D images and depth from a fixed set of viewpoints to generate 2D modality input. This stage harnesses insights from other multimodal data and learns enhanced 3D representations by aligning the features of the 3D modality with those of other multimodal data. Next, we continue training the 3D encoder using only point cloud data with improved masked point modeling. The model is tasked with predicting both global embeddings and token-wise embeddings from the first stage based on unmasked input point clouds. We also apply random shuffling to the encoder's output tokens before passing them to the decoder to avoid learning shortcuts. By focusing solely on point cloud data, this stage emphasizes local geometric and spatial relationships while retaining the semantic knowledge learned in the first stage. Our approach is model-agnostic and can be used with any MAE based point cloud architecture such as Point-MAE (Pang et al., 2022), Point-M2AE (Zhang et al., 2022a) etc. We conduct extensive experimental study on a broader range of 3D understanding tasks across five benchmark datasets that includes 3D object recognition, 3D semantic and part segmentation and 3D object detection. We also provide comprehensive ablation study to verify the effectiveness of different components of our approach. The main contributions of our paper are as follows:

- We propose a two-stage pre-training approach that learns semantically enriched 3D representations with high generalization capability. The first stage leverages knowledge from multiple foundation models using multimodal data, while the second stage enhances masked point modeling through local-global feature distillation.

- Our approach consistently outperforms competing methods on wide variety of 3D tasks. For example, in zero-shot setting, our first-stage pre-training surpasses ULIP2 (Xue et al., 2024) by $+1.2\%$ on ModelNet40 dataset (Wu et al., 2015). More notably, on the real-world ScanObjectNN dataset Uy et al. (2019), our approach significantly outperforms the Point-MAE baseline by $+5.18\%$, $+5.44\%$, and $+4.3\%$ on all three variants, respectively.

- We also demonstrate that our approach is compatible with any MAE-based architecture. When applied to Point-M2AE (Zhang et al., 2022a), a multi-scale hierarchical point cloud encoding framework, our

method achieves state-of-the-art (SOTA) performance across a range of downstream 3D understanding tasks.

## 2 Related Work

**Pre-training for Point Clouds.** There has been significant effort in developing pre-training methods for point clouds, following the recent success of self-supervised approaches in 2D vision and the NLP domain. Generally, 3D pre-training approaches can be divided into two categories: contrastive learning (Afham et al., 2022; Xie et al., 2020; Sanghi, 2020; Huang et al., 2021; Liu et al., 2022a) and generative learning (Yu et al., 2022; Pang et al., 2022; Zhang et al., 2022a; Zha et al., 2023a; Zhang et al., 2023). Contrastive learning aims to learn instance-discriminative representations by maximizing the similarity of different views of the same sample while minimizing the similarity of views from different samples. These views are typically generated using data augmentations. In 3D, PointContrast (Xie et al., 2020) uses geometric transformations to generate multiple views for learning discriminative representations. In contrast, generative pre-training focuses on learning representations by masking portions of the input and reconstructing them either in the input space or the latent space. Point-BERT (Yu et al., 2022), inspired by BERT (Devlin et al., 2019), pre-trains a transformer model by predicting the masked point tokens using tokenizer with discrete Variational AutoEncoder (dVAE). Point-MAE (Pang et al., 2022) follows the MAE (He et al., 2022) paradigm to reconstruct masked point patches in the point space. Point-M2AE (Zhang et al., 2022a) introduces a hierarchical multi-scale transformer to pre-train the model using masked point modeling. Point-FEMAE (Zha et al., 2023a) proposes local enhancement modules and incorporate global random and local block masking to learn compact 3D representations. Pix4Point (Qian et al., 2024) differs from cross-modal contrastive/generative methods; it adapts image-pretrained transformers to point clouds via weight transfer. Mamba3D (Han et al., 2024) proposes a state-space backbone with local pooling to better capture local geometry, achieving high accuracy with improved efficiency. In contrast, our work focuses on transformer-based architectures, and our two-stage framework is orthogonal and can be combined with alternative backbones such as Mamba3D.

**Cross-Modal Pre-training.** In addition to contrastive and generative pre-training, there have been efforts to learn representations by acquiring knowledge from other modalities such image, text etc. In the realm of 3D, CrossPoint (Afham et al., 2022) employs contrastive learning to align point clouds with their rendered images while ensuring invariance to spatial transformations. PointCLIP (Zhang et al., 2022b) projects point clouds into depth maps and leverages CLIP (Radford et al., 2021) to bridge 2D image–text representations with 3D data. ACT (Dong et al., 2022) introduces masked point modeling with feature distillation. The target features are generated by pre-training a cross-modal autoencoder that acquires knowledge from other modalities by leveraging a pre-trained vision transformer. OpenShape (Liu et al., 2023a)introduces large-scale multimodal 3D pretraining with millions of shapes, leveraging image–text–point correspondences to improve generalization across 3D recognition tasks. Uni3D (Zhou et al., 2024) introduces a scalable 3D foundation model by initializing a ViT from 2D pretraining and aligning point cloud features with image–text representations. By scaling up to one billion parameters, Uni3D achieves strong performance on zero-shot, few-shot, and open-world 3D tasks. ReCon (Qi et al., 2023) combines both contrastive and generative learning to incorporate knowledge from other modalities. ReCon++ (Qi et al., 2024) extends ReCon with stronger objectives and larger-scale training. I2P-MAE (Zhang et al., 2023) integrates 2D guided masking and 2D visual features, in addition to 3D coordinate reconstruction, by leveraging 2D pre-trained models. Point-Bind (Guo et al., 2023b) constructs a joint embedding space by aligning point clouds with image, text, video and audio via contrastive learning under ImageBind. It shows strong zero-shot, any-to-3D generation, and open-world understanding capabilities, further illustrating the power of multi-modal supervision in 3D learning. Multi-View masked learner (Chen et al., 2025) introduces a masked learner that projects point clouds into multi-view 2D features (using pose), with a two-stage teacher-student scheme and MSMH attention.

**Knowledge Distillation.** The concept of training a smaller network (student) from a large network (teacher) was first proposed for model compression (Bucilu et al., 2006). The goal is to transfer the "dark knowledge" from the teacher model to the student model. (Hinton et al., 2015) extends this idea for deep neural network by using the logits of the teacher model to distill knowledge in the student model. Following this breakthrough, a plethora of work (Furlanello et al., 2018; Cho & Hariharan, 2019; Mirzadeh et al.,

2019; Yang et al., 2018) have expanded on the use of logits or soft labels of the teacher model to guide the student model. Another line of work in this area is the feature distillation (Heo et al., 2019b; Huang & Wang, 2019; Heo et al., 2019a; Park et al., 2019; Kim et al., 2018; Peng et al., 2019), which focuses on transferring the features of the teacher model directly into the backbone of the student model. In 3D understanding, ACT (Dong et al., 2022) uses a cross-modal teacher as the target for masked point modeling in the 3D student model. The cross-modal teacher is trained to acquire knowledge from other modalities through self-supervised prompt tuning.

## 3 Comparison with Prior Two-Stage and Multimodal Methods

We summarize key distinctions between our approach and prior two-stage, distillation-based, and multimodal pretraining methods. Our framework decomposes pretraining into two stages: multi-teacher multimodal alignment (Stage-1) and geometry-focused masked point modeling via cross-stage feature distillation (Stage-2), enabling a clear separation between semantic alignment and geometric refinement.

Compared to ULIP Xue et al. (2023) and ULIP-2 Xue et al. (2024), which rely on a single CLIP teacher for global semantic alignment, we combine CLIP and DINOv2 to provide complementary semantic and dense spatial supervision. Our ablation studies show that neither teacher alone matches their combination, indicating that multi-teacher alignment produces richer representations that are critical for the subsequent distillation stage.

Compared to I2P-MAE Zhang et al. (2023), which integrates multimodal supervision directly into reconstruction objectives, our approach explicitly decouples multimodal alignment from masked point modeling. We first learn a semantic 3D encoder via multimodal alignment and then distill its representations during geometry-focused masked modeling. This separation avoids reconstruction-specific biases and enables compatibility with different MAE-based 3D encoders.

Compared to ACT Dong et al. (2022), which distills features from a cross-modal autoencoding teacher within a tightly coupled pipeline and primarily supervises masked tokens, our method uses a standalone Stage-1 encoder trained via multimodal alignment and applies both global and token-level supervision. We further introduce token shuffling to mitigate shortcut learning from visible tokens. Another key distinction lies in architectural generality. While ACT allows variation in the pretrained 2D backbone, its 3D teacher–student pipeline remains tied to a specific autoencoding architecture and integrates a single pretrained model at a time, limiting extensibility. In contrast, our two-stage formulation treats the Stage-1 3D encoder as a standalone semantic teacher and applies masked distillation in Stage-2 without modifying the encoder design. As a result, our framework is backbone-agnostic and can be directly applied to different MAE-based architectures, including Point-MAE, Point-M2AE, and 3DETR-based models, as well as SSM-based architectures such as PointMamba Liang et al. (2024), as demonstrated across multiple backbones and tasks. Our contribution is therefore a general pretraining framework rather than a new backbone.

Compared to ReCon Qi et al. (2023), which combines contrastive and generative objectives within a unified framework using a single vision-language teacher, our approach separates multimodal alignment and geometric modeling into two stages and employs multi-teacher supervision. A more detailed comparison with ReCon is provided in Appendix D.

Finally, LiDAR-based methods such as SLidR Sautier et al. (2022), Seal Liu et al. (2023b), SuperFlow Xu et al. (2024), and LargeAD Kong et al. (2026) primarily rely on spatial and temporal contrastive signals from sequential LiDAR data. In contrast, our framework leverages multimodal supervision from vision foundation models and cross-stage feature distillation, targeting a different supervision paradigm. These approaches are complementary rather than directly comparable.

## 4 Approach

Our goal is to learn semantically and geometrically enriched 3D representations from multiple modalities, such as images, depth maps, and text, by leveraging pre-trained vision and language models. We propose a simple yet effective two-stage pre-training approach that captures both multimodal 3D representations and

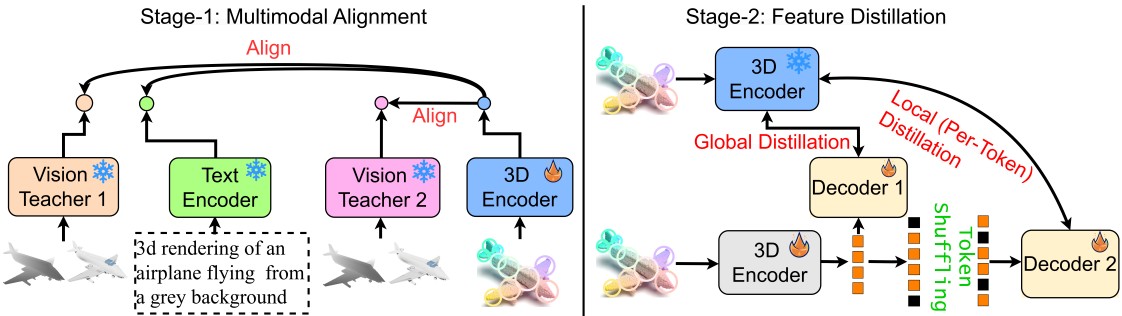

Figure 1: **Overview of our two-stage pre-training.** In the first stage, we pre-train the 3D encoder by aligning multimodal features with point cloud features using various vision and language foundation models. In the second stage, we further pre-train the 3D encoder with enhanced masked point modeling. The Stage-1 3D encoder generates global and token-level embeddings, which then serve as targets for the second-stage distillation.

local geometric features. Given a batch of multimodal data, we first pre-train the 3D encoder by aligning these modalities with point clouds using contrastive learning with vision and language pre-trained models. In the second stage, we fine-tune the 3D encoder solely on point clouds with enhanced masked point modeling, effectively distilling the knowledge acquired in the first stage. The 3D encoder from the first stage serves as a cross-modal teacher during this process. Figure 1 illustrates the pre-training pipeline of our approach.

## 4.1 Stage-1: Multimodal Representation Alignment

In the first stage, we conduct contrastive learning to align images, depth and text with point clouds. Specifically, given a point cloud $\mathcal{P} = \{\mathbf{p}_i | i = 1, 2, \ldots, N\} \in \mathbb{R}^{N \times 3}$ with $N$ coordinates, a randomly sampled 2D rendered image or associated depth $\mathcal{I}$, and a language description $\mathcal{T}$ (e.g., a description generated by BLIP-2), we extract modality-specific features using the modality-specific encoders. We pass the point clouds $\mathcal{P}$ to 3D point cloud encoder $E_{\mathrm{P}}$ to extract 3D representations $\mathbf{f}^{\mathrm{P}} = E_{\mathrm{P}}(\mathcal{P})$. Next, we extract text features $\mathbf{f}^{\mathrm{T}} = E_{\mathrm{T}}(\mathcal{T})$ using text encoder $E_{\mathrm{T}}$ and image features $\mathbf{f}^{\mathrm{I}} = \{\mathbf{f}^{(\mathrm{I,m})} | m = 1, 2, \ldots, N\}$ where $\mathbf{f}_m^{\mathrm{I}} = E_m^{\mathrm{I}}(\mathcal{I})$ using N different vision foundation models $E_m^{\mathrm{I}}$. In experiments, we use two vision foundation models i.e., $N = 2$. To match the feature dimension of the teacher model, we employ a teacher-specific projection layer. Following prior works (Xue et al., 2023; 2024; Radford et al., 2021), we apply a symmetric cross-entropy loss over the similarity score for multimodal alignment. More specifically, 3D-to-text alignment can be formulated as:

$$\mathcal{L}^{P \leftrightarrow T} = -\frac{1}{2} \sum_i \Big( \log \frac{\exp(\mathbf{f}_i^{\mathrm{P}} \mathbf{f}_i^{\mathrm{T}}/\tau)}{\sum_j \exp(\mathbf{f}_i^{\mathrm{P}} \mathbf{f}_j^{\mathrm{T}}/\tau)} + \log \frac{\exp(\mathbf{f}_i^{\mathrm{T}} \mathbf{f}_i^{\mathrm{P}}/\tau)}{\sum_j \exp(\mathbf{f}_i^{\mathrm{T}} \mathbf{f}_j^{\mathrm{P}}/\tau)} \Big). \tag{1}$$

Similarly, the 3D-to-image alignment for a vision teacher $m$ can be written as:

$$\mathcal{L}_m^{P \leftrightarrow I} = -\frac{1}{2} \sum_i \Big( \log \frac{\exp(\mathbf{f}_i^{\mathrm{P}} \mathbf{f}_i^{(\mathrm{I,m})}/\tau)}{\sum_j \exp(\mathbf{f}_i^{\mathrm{P}} \mathbf{f}_j^{(\mathrm{I,m})}/\tau)} + \log \frac{\exp(\mathbf{f}_i^{(\mathrm{I,m})} \mathbf{f}_i^{\mathrm{P}}/\tau)}{\sum_j \exp(\mathbf{f}_i^{(\mathrm{I,m})} \mathbf{f}_j^{\mathrm{P}}/\tau)} \Big), \tag{2}$$

Overall, the pre-training objective for stage-1 is to minimize the sum of two contrastive alignment losses, which is:

$$\mathcal{L}_{\text{stage1}} = \mathcal{L}^{P \leftrightarrow T} + \sum_m \mathcal{L}_m^{P \leftrightarrow I}. \tag{3}$$

Although Stage-1 follows the multimodal alignment paradigm proposed in recent works (Xue et al., 2023; 2024; Guo et al., 2023b), our key contribution lies in extending this paradigm through multi-teacher supervision using multiple vision foundation models, and integrating it into a unified two-stage framework. In this design, Stage-1 learns a semantically rich 3D encoder that serves as the foundation for the subsequent geometry-focused masked point modeling in Stage-2.

**Multi-Teacher Multimodal Alignment.** While multimodal contrastive learning is well established in 3D representation learning, prior approaches typically rely on a single teacher (e.g., CLIP), which primarily captures global semantic alignment. In contrast, we employ multi-teacher supervision by combining CLIP and DINOv2, which provide complementary signals: CLIP encodes high-level semantic information through image-text pretraining, while DINOv2 captures dense spatial features from image-only supervision.

Our ablation studies (Table 11 and Figure 3 in the Appendix C) show that neither teacher alone matches the performance of their combination across both zero-shot and fine-tuning benchmarks. Importantly, the role of multi-teacher alignment in our framework is not to ensemble predictions at inference time, but to inject complementary semantic and spatial cues into the 3D encoder during pretraining. This results in richer representations that are crucial for the subsequent geometry-focused distillation stage, where single-teacher alignment yields consistently weaker performance.

### 4.2 Stage-2: Masked Point Modeling

While the first stage captures semantic relationships through multimodal alignment, pre-training on multimodal data using contrastive learning may not fully capture the complex nature of the 3D data, as it tends to overlook local geometric features and spatial relationships, focusing primarily only on the global similarities between modalities. To address this, our second-stage pre-training focuses on learning representations solely from point cloud data. We build upon the masked point modeling techniques (Dong et al., 2022; Zhang et al., 2022a), while ensuring that the model retains the semantics and other representations learned in the first stage.

In this stage, we initialize the second-stage 3D encoder with the first-stage model's weights and then pre-train it on point cloud data using an enhanced masked point modeling approach. To prevent the model from learning shortcuts, we incorporate a token shuffling scheme. Additionally, we introduce both global and local token-wise feature distillation strategies, which allow the second-stage model to effectively leverage the knowledge acquired during the first stage. Specifically, the second-stage model learns to predict all tokens from the first-stage model based on the masked point cloud.

**Local Distillation.** Since we initialized the model from the first stage, it may begin to develop shortcuts for decoding the unmasked tokens while predicting the masked ones. This could simplify the task, making it easier than predicting all the tokens simultaneously. To mitigate this issue, we first concatenate the visible and masked tokens, randomly shuffle the token sequence, and then pass it to the decoder, where positional embeddings are added after the shuffling. This operation encourages the model to avoid learning shortcuts for unmasked tokens. The approach is analogous to the Jigsaw pretext task (Noroozi & Favaro, 2017) in SSL, which the decoder tries to solve for the unmasked tokens while predicting the masked ones. This masked point modeling, referred to as local feature distillation, minimizes the negative cosine similarity between the output tokens of the first-stage and second-stage models.

$$\mathcal{L}_{local} = -\sum_{i=1}^{N_t} \mathbf{1} - \frac{\mathbf{z}_i^1 \cdot \mathbf{z}_i^2}{\|\mathbf{z}_i^1\| \cdot \|\mathbf{z}_i^2\|} \tag{4}$$

where $N_t$ is the total number of tokens, and $\mathbf{z}_i^1$ and $\mathbf{z}_i^2$ represents the $i^{th}$ token from the first-stage model and the decoder-2 of the second-stage model, respectively.

Table 1: **Few-shot classification results on ModelNet40.** Overall mean accuracy (%) with standard deviation (without voting) is reported.

| Methods | 5-way | | 10-way | |
|---|---|---|---|---|
| | 10-shot | 20-shot | 10-shot | 20-shot |
| *Supervised Learning Only* | | | | |
| PointNet Qi et al. (2017a) | 52.0±3.8 | 57.8±4.9 | 46.6±4.3 | 35.2±4.8 |
| DGCNN Wang et al. (2019) | 31.6±2.8 | 40.8±4.6 | 19.9±2.1 | 16.9±1.5 |
| OcCo Wang et al. (2021) | 90.6±2.8 | 92.5±1.9 | 82.9±1.3 | 86.5±2.2 |
| *with Single-Modal Self-Supervised Representation Learning* | | | | |
| Point-BERT Yu et al. (2022) | 94.6±3.1 | 96.3±2.7 | 91.0±5.4 | 92.7±5.1 |
| MaskPoint Liu et al. (2022b) | 95.0±3.7 | 97.2±1.7 | 91.4±4.0 | 93.4±3.5 |
| Point-MAE Pang et al. (2022) | 96.3±2.5 | 97.8±1.8 | 92.6±4.1 | 95.0±3.0 |
| Point-M2AE Zhang et al. (2022a) | 96.8±1.8 | 98.3±1.4 | 92.3±4.5 | 95.0±3.0 |
| PointGPT Chen et al. (2024) | 96.8±2.0 | 98.6±1.1 | 92.6±4.6 | 95.2±3.4 |
| Point-FEMAE Zha et al. (2023a) | 97.2±1.9 | 98.6±1.3 | 94.0±3.3 | 95.8±2.8 |
| *with Cross-Modal Self-Supervised Representation Learning* | | | | |
| ACT Dong et al. (2022) | 96.8±2.3 | 98.0±1.4 | 93.3±4.0 | 95.6±2.8 |
| Joint-MAE Guo et al. (2023a) | 96.7±2.2 | 97.9±1.9 | 92.6±3.7 | 95.1±2.6 |
| I2P-MAE Zhang et al. (2023) | 97.0±1.8 | 98.3±1.3 | 92.6±5.0 | 95.5±3.0 |
| TAP Wang et al. (2023) | 97.3±1.8 | 97.8±1.9 | 93.1±2.6 | 95.8±1.0 |
| ReCon Qi et al. (2023) | 97.3±1.9 | 98.9±1.2 | 93.3±3.9 | 95.8±3.0 |
| **Ours (Point-M2AE)** | **97.6±1.7** | **99.0±0.6** | **94.1±3.0** | **96.2±2.6** |
| *Improve (over Point-M2AE)* | +0.8 | +0.7 | +1.8 | +1.2 |
| Mamba3D Han et al. (2024) | 96.4±2.2 | 98.2±1.2 | 92.4±4.1 | 95.2±2.9 |

**Global Distillation.** We find that our second-stage model slightly under-performs compared to the first-stage model on synthetic data when using only local distillation. This can likely be attributed that to catastrophic forgetting (McCloskey & Cohen, 1989) during the two-stage pre-training. To address this issue, we distill the global 3D representations from the first stage using the full point cloud. Given the visible tokens from the second-stage 3D encoder, we employ a decoder network (e.g., transformer blocks or MLP) to output a global embedding. We do not apply token shuffling or add positional embedding for this decoder. We then align the second-stage global embedding with the global 3D embedding from the first stage by minimizing the negative cosine similarity.

$$\mathcal{L}_{global} = -\sum_i \mathbf{1} - \frac{\mathbf{f}^{\mathrm{P}}_{(1,i)} \cdot \mathbf{f}^{\mathrm{P}}_{(2,i)}}{\|\mathbf{f}^{\mathrm{P}}_{(1,i)}\| \cdot \|\mathbf{f}^{\mathrm{P}}_{(2,i)}\|} \tag{5}$$

where $\mathbf{f}^{\mathrm{P}}_{(1,i)}$ and $\mathbf{f}^{\mathrm{P}}_{(2,i)}$ represents the global embeddings from the first-stage model and the second-stage model respectively.

Overall, the pre-training objective of the second stage is the sum of two feature distillation losses, given by:

$$\mathcal{L}_{\mathrm{stage2}} = \mathcal{L}_{local} + \mathcal{L}_{global}. \tag{6}$$

# 5 Experiments

We evaluate our method across a range of 3D understanding tasks, including object classification, indoor scene segmentation, part segmentation, and 3D object detection, to assess its generalization capability. Due to space constraints, detailed results on segmentation and detection tasks are provided in Appendix B. Implementation details are also included in the Appendix A.

Table 2: **Zero-shot 3D classification on ModelNet40.** We report top-1 and top-5 accuracy. A "✓" in the Manual captions means that the models use text prompt with category names to generate language description for the 3D data while "✗" means the opposite.

| Model | Pre-train dataset | Pre-train method | Manual captions? | ModelNet40 top-1 | top-5 |
|---|---|---|---|---|---|
| PointCLIP (Zhang et al., 2022b) | – | – | – | 19.3 | 34.8 |
| PointCLIPv2 (Zhu et al., 2022) | – | – | – | 63.6 | 85.0 |
| ReCon (Qi et al., 2023) | ShapeNet | ReCon (Qi et al., 2023) | ✓ | 61.2 | 78.1 |
| CLIP2Point (Huang et al., 2023b) | ShapeNet | CLIP2Point (Huang et al., 2023b) | ✗ | 49.5 | 81.2 |
| Point-BERT (Yu et al., 2022) | ShapeNet | OpenShape (Liu et al., 2023a) | ✓ | 70.3 | 91.3 |
| Point-BERT (Yu et al., 2022) | ShapeNet | ULIP Xue et al. (2023) | ✓ | 60.4 | 84.0 |
| | | Ours | ✓ | **65.8** | **93.5** |
| | | ULIP-2 Xue et al. (2024) | ✗ | 70.0 | 89.7 |
| | | Ours | ✗ | **71.2** | **93.0** |
| | Objaverse(no LVIS) + ShapeNet | ULIP Xue et al. (2023) | ✓ | 68.6 | 86.4 |
| | Objaverse + ShapeNet | ULIP Xue et al. (2023) | ✓ | 69.6 | 85.9 |
| PointM2AE (Zhang et al., 2022a) | ShapeNet | Ours | ✗ | **73.4** | **94.5** |

## 5.1 Results on 3D Object Classification

We first evaluate the performance of our approach on 3D object classification. Both the Point-MAE and Point-M2AE based 3D encoders are used for fine-tuning. Following previous works (Qi et al., 2023; Dong et al., 2022), we adopt the same transfer learning protocols: Full, MLP-3, and MLP.

**3D Synthetic Object Recognition** We evaluate our approach on ModelNet40 (Wu et al., 2015), which contains 12311 3D CAD objects spanning 40 categories. During fine-tuning, we apply the *Scale and Translate* data augmentation. The results are reported in Table 3, both with and without the voting strategy. It can be seen from the table that our approach sets the state-of-art for these strategies. Our pre-trained model, using multi-scale architecture (Zhang et al., 2022a), achieves **94.3%** accuracy under full protocol, improving by **+0.9** over Point-M2AE performance. Using the standard plain transformer, our approach achieves **94.0%** accuracy, improving by **+0.8** over Point-MAE baseline. Furthermore, compared to ACT (Dong et al., 2022), a two-stage pre-training approach, our method outperforms it by **+0.8%** and **+1.1%** on both strategies, respectively. Finally, for other transfer learning protocols, our approach consistently outperforms the other baselines, including ReCon (Qi et al., 2023).

**3D Real-World Object Recognition** Next, we evaluate our approach on ScanObjectNN (Uy et al., 2019) dataset, which consists of 15k 3D objects. Following previous works (Qi et al., 2023; Dong et al., 2022), we evaluate on three variants of this dataset: OBJ-BG, OBJ-ONLY, and PB-T50-RS. We only apply *rotation* augmentation during fine-tuning as done in (Qi et al., 2023; Dong et al., 2022). The results are reported in the Table 3 using both standard transformer and multi-scale transformer architectures. As shown in the table, our approach sets new state-of-the results across all variants of ScanObjectNN. More specifically, the standard transformer fine-tuned model improves by **5.18%**, **5.44%**, and **4.92%** over Point-MAE across all three variants, respectively. Similarly, compared to Point-M2AE, our approach shows improvements of **4.88%**, **5.41%**, and **4.77%** on these variants. Furthermore, compared to ReCon (Qi et al., 2023), our approach show improvement of **+0.9%** on OBJ_BG variant of ScanObjectNN. We further evaluate the generalization and compatibility of our framework across architectures and fine-tuning strategies. First, we apply our method to PointMamba Liang et al. (2024), an SSM-based backbone, and observe consistent improvements over the single-modal baseline, demonstrating that the proposed framework is not restricted to transformer-based encoders. Second, we evaluate PointGST Liang et al. (2025) on top of our pretrained models and find that it provides additional gains, indicating that our learned representations are complementary to parameter-efficient fine-tuning methods. Furthermore, under the MLP-3 protocol, our approach with the standard transformer model outperforms ReCon by **+1.08%** on the ScanObjectNN hard variant, demonstrating that the frozen features are more representative and discriminative compared to other SSL methods. This observation holds consistent across the other two variants. Lastly, we can draw the same conclusion for MLP protocol.

Table 3: **Classification accuracy (%) on ScanObjectNN and ModelNet40.** We report the overall accuracy (%) on three variants of ScanObjectNN. For ModelNet40, we report the overall accuracy (%) for both with and without voting. "#P" means the model's parameters. Results with "+ PointGST" are obtained by applying parameter-efficient fine-tuning on top of the corresponding pretrained backbone.

| Methods | #P | ScanObjectNN | | | | ModelNet40 | | |
|---|---|---|---|---|---|---|---|---|
| | | Input | OBJ_BG | OBJ_ONLY | PB_T50_RS | Input | w/o Vote | w/ Vote |
| *Supervised Learning Only* | | | | | | | | |
| PointNet Qi et al. (2017a) | 3.5 | 1k Points | 73.3 | 79.2 | 68.0 | 1k Points | 89.2 | - |
| PointNet++ Qi et al. (2017b) | 1.5 | 1k Points | 82.3 | 84.3 | 77.9 | 1k Points | 90.7 | - |
| DGCNN Wang et al. (2019) | 1.8 | 1k Points | 82.8 | 86.2 | 78.1 | 1k Points | 92.9 | - |
| SimpleView Goyal et al. (2021) | - | 6 Images | - | - | 80.5±0.3 | 6 Images | 93.9 | - |
| MVTN Hamdi et al. (2021) | 11.2 | 20 Images | 92.6 | 92.3 | 82.8 | 12 Images | 93.8 | - |
| PointMLP Ma et al. (2022) | 12.6 | 1k Points | - | - | 85.4±0.3 | 1k Points | 94.5 | - |
| SFR Zha et al. (2023b) | - | 20 Images | - | - | 87.8 | 20 Images | 93.9 | - |
| P2P-HorNet Wang & Yoon (2021) | 195.8 | 40 Images | - | - | 89.3 | 40 Images | 94.0 | - |
| *with Single-Modal Self-Supervised Learning* | | | | | | | | |
| Point-BERT Yu et al. (2022) | 22.1 | 1k Points | 87.43 | 88.12 | 83.07 | 1k Points | 92.7 | 93.2 |
| MaskPoint Liu et al. (2022b) | - | 2k Points | 89.30 | 88.10 | 84.30 | 1k Points | - | 93.8 |
| Point-MAE Pang et al. (2022) | 22.1 | 2k Points | 90.02 | 88.29 | 85.18 | 1k Points | 93.2 | 93.8 |
| Point-M2AE Zhang et al. (2022a) | 15.3 | 2k Points | 91.22 | 88.81 | 86.43 | 1k Points | 93.4 | 94.0 |
| PointMamba Liang et al. (2024) | 12.3 | 2k Points | 94.32 | 92.60 | 89.31 | - | - | - |
| PointGPT Chen et al. (2024) | 19.5 | 2k Points | 91.60 | 90.00 | 86.90 | 1k Points | - | 94.0 |
| Point-FEMAE Zha et al. (2023a) | 27.4 | 2k Points | 95.18 | 93.29 | 90.22 | 1k Points | 94.0 | 94.5 |
| *with Cross-Modal Self-Supervised Learning* | | | | | | | | |
| ACT Dong et al. (2022) | 22.1 | 2k Points | 93.29 | 91.91 | 88.21 | 1k Points | 93.2 | 93.7 |
| Joint-MAE Guo et al. (2023a) | - | 2k Points | 90.94 | 88.86 | 86.07 | 1k Points | - | 94.0 |
| I2P-MAE Zhang et al. (2023) | 15.3 | 2k Points | 94.14 | 91.57 | 90.11 | 1k Points | 93.7 | 94.1 |
| TAP Wang et al. (2023) | 22.1 | 2k Points | 90.36 | 89.50 | 85.67 | - | - | - |
| ReCon Qi et al. (2023) | 43.6 | 2k Points | 95.18 | 93.63 | 90.63 | 1k Points | 94.1 | 94.5 |
| ReCon + PointGST Liang et al. (2025) | 44.2 | 2k Points | 94.49 | 92.94 | 89.49 | - | - | - |
| PViT Qian et al. (2024) | - | - | - | - | 85.7 | - | - | - |
| PViT+Pix4Point Qian et al. (2024) | - | - | - | - | 87.9 | - | - | - |
| Multi-View ML (Point-MAE) Chen et al. (2025) | 22.1 | - | 93.32 | 92.69 | 88.93 | - | 93.8 | 94.1 |
| Multi-View ML (Point-M2AE) Chen et al. (2025) | 22.1 | - | 95.10 | 93.56 | 90.37 | - | 94.0 | 94.4 |
| **Ours (Point-MAE)** | 22.1 | 2k Points | 95.20 | 93.73 | 90.10 | 1k Points | 94.0 | 94.5 |
| *Improve (over Point-MAE)* | | | +5.18 | +5.44 | +4.92 | - | +0.8 | +0.7 |
| Ours (Point-MAE + PointGST Liang et al. (2025)) | 22.7 | 2k Points | 95.36 | 94.11 | 90.18 | - | - | - |
| **Ours (Point-M2AE)** | 15.3 | 2k Points | **96.10** | **94.25** | **91.20** | 1k Points | **94.3** | **94.6** |
| *Improve (over Point-M2AE)* | - | - | +4.88 | +5.41 | +4.77 | - | +0.9 | +0.6 |
| **Ours (PointMamba)** | 12.1 | 2k Points | 95.15 | 93.19 | 90.11 | - | - | - |
| *Improve (over PointMamba)* | - | - | +0.83 | +0.59 | +0.80 | - | - | - |
| *Large-Scale Models (Reference)* | | | | | | | | |
| PointGPT-L + PointGST Liang et al. (2025) | 362.9 | 2k Points | 98.97 | 97.59 | 94.83 | - | - | - |
| *with Self-Supervised Representation Learning (MLP-Linear)* | | | | | | | | |
| Point-MAE Pang et al. (2022) | 22.1 | 2k Points | 82.77±0.30 | 83.23±0.16 | 74.13±0.21 | 1k Points | 91.22±0.26 | - |
| ACT Dong et al. (2022) | 22.1 | 2k Points | 85.20±0.83 | 85.84±0.15 | 76.31±0.26 | 1k Points | 91.36±0.17 | - |
| ReCon Qi et al. (2023) | 43.6 | 2k Points | 89.50±0.20 | 89.72±0.17 | 81.36±0.14 | 1k Points | 92.47±0.22 | - |
| **Ours (Point-MAE)** | 22.1 | 2k Points | **90.57±0.19** | **90.68±0.14** | **82.41±0.16** | 1k Points | **92.68±0.10** | - |
| *with Self-Supervised Representation Learning (MLP-3)* | | | | | | | | |
| Point-MAE Pang et al. (2022) | 22.1 | 2k Points | 84.29±0.55 | 85.24±0.67 | 77.34±0.12 | 1k Points | 92.33±0.09 | - |
| ACT Dong et al. (2022) | 22.1 | 2k Points | 87.14±0.22 | 87.90±0.40 | 81.52±0.19 | 1k Points | 92.69±0.18 | - |
| ReCon Qi et al. (2023) | 43.6 | 2k Points | 90.62±0.22 | 90.71±0.30 | 83.80±0.42 | 1k Points | 93.00±0.10 | - |
| **Ours (Point-MAE)** | 22.1 | 2k Points | **91.56±0.33** | **91.38±0.40** | **84.88±0.48** | 1k Points | **93.35±0.17** | - |

**Few-shot Classification**  We opt for ModelNet40 for few-shot classification experiments, following previous works (Dong et al., 2022; Qi et al., 2023; Pang et al., 2022) in 3D pre-training. We use the standard "n-way", "m-shot" configuration, where "n" represents the number of randomly sampled class and "m" represents the number of training examples for each class. We report the mean accuracy with standard deviation by evaluating on 10 independent experiments. It can be seen from the Table 1 that our approach achieves the state-of-the-art (SOTA) performance on all four settings. More specifically, our method shows improvements of **+0.8%**, **+0.7%**, **+1.8%** and **+1.2%** over Point-M2AE, respectively. Furthermore, compared to ReCon and ACT, our model shows smaller deviations, suggesting that the pre-trained model has learned more robust and discriminative features that can be better adapted to downstream tasks in a low-data regime.

Table 4: **Ablation study for the proposed pre-training strategies.** Overall accuracy (%) without voting is reported.

| Global Distillation | Token Shuffling | Local Distillation | MN40 | ScanObjNN OBJ_BG |
|:---:|:---:|:---:|:---:|:---:|
| ✓ | ✓ | ✓ | **94.3** | **96.1** |
| ✗ | ✓ | ✓ | 94.1 | 95.8 |
| ✓ | ✗ | ✓ | 93.9 | 95.7 |
| ✗ | ✗ | ✓ | 93.6 | 95.5 |
| ✗ | ✗ | ✗ | 93.7 | 95.0 |

Table 5: **Training Costs**. We compare the training costs in terms pre-training GPU hours, fine-tuning GPU hours and number of parameters on NVIDIA A100.

| Method | #Params | Stage-1 Pre-train GPU hours | Stage-2 Pre-train GPU hours | Total Pre-train GPU hours | Fine-tune GPU hours | ScanObjectNN |
|:---|:---:|:---:|:---:|:---:|:---:|:---:|
| Point-MAE | 22.1M | - | 15.2h | 15.2h | 9h | 85.2 |
| Point-MAE + 3.5 x pre-training | 22.1M | - | 54h | 54h | 9h | 85.1 |
| Ours (Point-MAE) | 22.1M | 24h | 30h | 54h | 9h | **90.1** |

**Zero-shot 3D Object Classification**  Our stage-1 pre-training aligns different modalities with the point cloud, enabling us to assess whether our approach exhibits a strong zero-shot capability. Following previous works (Qi et al., 2023; Xue et al., 2023), we conduct experiments on ModelNet40 (Wu et al., 2015). The results are shown in Table 2. Following PointCLIP (Zhang et al., 2022b) and ULIP-2 (Xue et al., 2024), we use both prompt templates with category labels and the top-1 BLIP-2 (Li et al., 2023) captions as the textual descriptions of the 3D objects. From Table 2, we observe that: (i) Our approach significantly outperforms the other zero-shot approaches; (ii) Compared to ULIP (Xue et al., 2023) and ReCon (Qi et al., 2023), our approach with prompt templates surpasses them by a clear margin achieving **93.5%** top-5 accuracy; (iii) Point-M2AE based 3D encoder trained with top-1 BLIP-2 descriptions achieves **73.4%** top-1 accuracy, surpassing ULIP-2 by **3.4%**.

Table 6: **Segmentation Results on S3DIS Area 5.** We report mean accuracy and mean IoU across all categories, i.e., mAcc (%) and mIoU(%) respectively.

| Methods | Semantic Seg. | |
|:---|:---:|:---:|
| | mAcc | mIoU |
| PointNet Qi et al. (2017a) | 49.0 | 41.1 |
| PointNet++ Qi et al. (2017b) | 67.1 | 53.5 |
| *with Single-Modal Self-Supervised Representation Learning* | | |
| Transformer Vaswani et al. (2017) | 68.6 | 60.0 |
| Point-MAE Pang et al. (2022) | 69.9 | 60.8 |
| *with Cross-Modal Self-Supervised Representation Learning* | | |
| ACT Dong et al. (2022) | 71.1 | 61.2 |
| **Ours (Point-MAE)** | **72.3** | **62.9** |

## 5.2 Results on 3D Scene Segmentation

Next, we evaluate the approach on scene segmentation, a considerably complex task that requires the model to understand local geometric relationships and contextual semantics. We conduct experiment using

S3DIS (Armeni et al., 2016) and use Area5 for the evaluation. The results are reported in Table 6. Our approach outperforms both PointMAE and ACT by **+2.4%** and **+1.1%** in mAcc, respectively.

### 5.3 Ablation Study

**Effect of second stage pre-training objectives.** To verify the effectiveness of different components of our second stage pre-training, we conduct an ablation study comparing the performance of token shuffling and global distillation with full second-stage model on ModelNet40 and OBJ_BG variant of ScanObjectNN. The results are reported in Table 4. We observe that: i) Both global distillation and token shuffling improve the overall performance of the second-stage model; ii) Token shuffling, in particular, provides a greater performance gain compared to global distillation. We conjecture that the more challenging learning objective introduced by token shuffling improves the overall 3D representations by encouraging the second-stage model to better capture geometric features in the point clouds; iii) Global distillation also boosts the performance of the second-stage model and helps mitigate catastrophic forgetting, particularly on ModelNet40.

Table 7: **Comparison between local distillation loss and reconstruction in point cloud space on ModelNet40 dataset.** Left table report results using Point-MAE while right table reports results using Point-M2AE. Overall accuracy (%) without voting is reported.

| Target | Acc (%) |
|---|---|
| Point clouds | 93.5 |
| Features | **94.0** |

| Target | Acc (%) |
|---|---|
| Point clouds | 93.7 |
| Features | **94.3** |

**Reconstruction in the point space?** We conduct experiments to compare pre-training the model using masked point modeling in the point cloud space with proposed local distillation. In Table 7, we report the results on ModelNet40 using Point-MAE and Point-M2AE. We find that reconstructing in the point cloud space degrades the overall performance for both architectures. One plausible reason for this is that, given the initialization from the stage-1 weights, the reconstruction task becomes easier, allowing the model to quickly find local optima when combined with global distillation, which may consequently degrade downstream performance.

**Training Costs.** As shown in Table 5, our two-stage pretraining requires approximately 3.5× more computation than Point-MAE. However, this cost is incurred only during pretraining, while inference and fine-tuning complexity (in terms of FLOPs and parameter count) remain comparable to Point-MAE and related backbones. Despite the increased pretraining cost, our approach yields consistent gains across downstream tasks (e.g., +4–5% on ScanObjectNN). Importantly, simply extending the pretraining time of Point-MAE to match our training budget does not lead to similar improvements (Table 5), suggesting that the gains arise from the proposed training strategy rather than additional computation. Furthermore, our framework is model-agnostic and does not introduce any custom layers, making it compatible with existing architectures without increasing deployment cost.

**Stage-1 v/s Stage-2 results.** In the last row of Table 4, we present the performance of our stage-1 pre-training. We observe that the stage-1 pre-trained model performs competitively, outperforming both Point-MAE and Point-M2AE on ModelNet40 and ScanObjectNN. Furthermore, by incorporating the stage-2 pre-training components, our approach achieves the highest performance on both datasets.

**Additional Baselines** We compare our approach to two additional baselines. The first baseline follows a single-stage pre-training with a multi-task learning paradigm. The second baseline reverses the learning order, performing masked point modeling in Stage-1 and multi-modal alignment in Stage-2. The results are reported in Table 8. It can be seen that both the baselines perform worse as compared to our approach. Joint training can lead to conflicting gradients which degrades the overall performance when combining these objectives. Reversing the order would prioritize local geometry too early, making it harder to align noisy text

and captions effectively. Morever, random initialization in Stage-2 degrades token-wise distillation and yields lower accuracy.

Table 8: **Results on additional baselines.** Overall accuracy (%) without voting is reported.

| Method. | MN40 | ScanOBJNN |
|---|---|---|
| Joint training (single stage) | 92.2 | 82.78 |
| Stage-2 -> Stage-1 | 93.6 | 87.1 |
| Stage-2 (random init) | 93.7 | 88.2 |
| **Ours (Point-MAE)** | **94.0** | **90.1** |

# 6 Conclusion

In this paper, we present a novel two-stage pre-training approach for point clouds, leveraging foundation models and feature distillation. Our method tackles the scalability challenge by incorporating diverse language descriptions generated by multimodal models such as BLIP-2, alongside images and depth maps associated with point clouds. In the first stage, multiple foundation models are used to learn rich multimodal representations. In the second stage, the model is further pre-trained by improved masked point modeling, incorporating feature distillation and token shuffling. Our approach learns rich semantic and geometric representations, achieving state-of-the-art performance across a wide range of 3D understanding tasks. Furthermore, it is model-agnostic and can be applied to any transformer-based 3D encoder.

# 7 Discussion and Limitations

Our two-stage pretraining approach leverages multimodal supervision and foundation models to learn rich semantic and geometric representations, and demonstrates consistent effectiveness across multiple 3D downstream tasks. However, several limitations remain. In Stage-1, we rely on single-view images or depth maps, which may not fully capture the complex geometry of 3D objects. While we pretrain on ShapeNet, which contains approximately 55k objects, incorporating multi-view supervision could further improve geometric consistency. In addition, designing a dedicated depth encoder could enable better alignment between point clouds, images, and language representations.

Our pretraining is primarily conducted on ShapeNet, which is a synthetic dataset. Although our models generalize well to real-world datasets such as ScanObjectNN, S3DIS, and ScanNet, scaling to larger and more diverse real-world datasets (e.g., Objaverse-XL (Deitke et al., 2023)) remains an important direction for future work. Furthermore, our experiments focus on standard-sized backbones due to computational constraints, and we do not explicitly study scaling behavior with very large models or datasets. Extending the framework to larger-scale architectures and datasets is a promising direction for future work.

While we demonstrate that our framework generalizes across multiple architectures, including an SSM-based backbone (PointMamba), our evaluation is primarily focused on object-centric and indoor scene understanding tasks. Extending the framework to large-scale outdoor LiDAR benchmarks (e.g., KITTI Geiger et al. (2013) or Waymo Sun et al. (2020)) would require addressing domain-specific challenges such as long-range sparsity and temporal aggregation, and is left for future work.

The multimodal supervision in Stage-1 relies on BLIP-2–generated captions with fixed prompt templates, which provide a scalable but limited semantic signal. Incorporating more diverse or human-generated captions could further improve representation quality and semantic richness. Finally, our two-stage training pipeline introduces additional computational cost compared to single-stage methods. In addition, the fixed multi-view rendering strategy used in Stage-1 may not fully capture occluded regions of 3D objects. Exploring more efficient training strategies, as well as adaptive or multi-view rendering approaches, are important directions for future work.

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
