# A   Implementation Details

In this section, we present the implementation details for pre-training and fine-tuning. All the experiments are done using NVIDIA A100 GPUs. We use ShapeNet (Chang et al., 2015) as the pre-training dataset, which consists of 51,300 3D objects spanning over 55 categories. Following ULIP (Xue et al., 2023), we sample 12 rendered images and depth maps that are 30 degrees apart. For each rendered image, we use BLIP-2 (Li et al., 2023) to generate detailed descriptions, as described in ULIP-2 (Xue et al., 2024). We select the top-1 description based on image-text similarity to represent the language modality. During stage-1 pre-training, we employ variants of CLIP (Radford et al., 2021) and DINOv2 (Oquab et al., 2023) as vision teachers, and use the text encoder from CLIP (Radford et al., 2021) as the text teacher. These models remain frozen throughout this stage. For pre-training, we use two architectures (i) Point-MAE (Pang et al., 2022), which features a standard transformer-based encoder that consists of 12 blocks and a decoder with 4 transformer blocks. The number of heads is 6, and the hidden dimension of the blocks is 384. (ii) Point-M2AE (Zhang et al., 2022a), which employs a multi-scale pyramid architecture that consists of a 3-stage encoder and a 2-stage decoder. Each stage in the encoder has 5 blocks, while the decoder has only 1 block per stage. We pre-train the models on the train set of ShapeNet dataset which consists of 41,952 3D shapes.

## A.1   Pre-training Setup

**ShapeNetCore.**   We use ShapeNetCore (Chang et al., 2015) to pre-train the models using our two-stage approach. In stage 1, we pre-train the models for 300 epochs with a batch size of 128 and a learning rate of 5e-4 on 4 GPUs. The weight decay is set to 5e-2, and we use AdamW (Loshchilov & Hutter, 2017) as the optimizer. The image resolution is set to 224x224. For data augmentation, we apply random point dropout, random scaling, random shifting, random perturbation, and random rotation following ULIP (Xue et al., 2023). We sample 8192 points from each 3D object. In stage 2, we pre-train the models for another 300 epochs with a batch size of 128 and a learning rate of 5e-4 on a single GPU. The weight decay remains 5e-2, and AdamW is used as the optimizer. For data augmentation, we apply random scaling and translation, and we sample 1024 points from each 3D shape. For the multi-scale hierarchical architecture, we sample 2048 points following Point-M2AE (Zhang et al., 2022a). We adopt a cosine learning rate scheduler with 10 warm-up epochs during both stages of pre-training. The main results in the tables are reported using CLIP-B and DINOv2-B, unless otherwise specified.

**ScanNetv2.**   For 3D object detection, we pre-train the models using ScanNetv2 (Dai et al., 2017), which consists of approximately 2.5M RGB-D scans from 1513 indoor scenes. Following ACT (Dong et al., 2022) and MaskPoint (Liu et al., 2022b), we use the ScanNet-Medium subset, which consists of approximately 25K frames sampled every 10th frame. We adapt the same encoder from 3DTER and 3DTER-m, and the same decoder for local distillation used by ACT and Point-M2AE. In stage 1, we pre-train the encoders for 100 epochs with a learning rate of 5e-4. In stage 2, we further pre-train the 3DTER-based encoder for 300 epochs and the 3DTER-m-based encoder for 1080 epochs, following the settings from Point-M2AE (Zhang et al., 2022a). For other pre-training details, we follow the ShapeNet pre-training procedure.

## A.2   Fine-tuning Setup

All the fine-tuning experiments are conducted on a single GPU.

**ModelNet40.**   ModelNet40 (Wu et al., 2015) consists of approximately 12k 3D synthetic objects, with 9,843 samples used for training and the remaining 2,468 used for validation. For the standard vanilla transformer, we fine-tune the model for 300 epochs with a batch size of 32 and a learning rate of 1e-5. The weight decay is set to 0.05, and we use AdamW as the optimizer along with a cosine learning rate scheduler. We sample 1024 points from each 3D shape and apply random scaling and translation augmentations. For the Point-M2AE-based 3D encoder, we maintain the same settings, except that we use a learning rate of 5e-4. For few-shot classification experiments, we fine-tune the model for 150 epochs.

**ScanObjectNN.** ScanObjectNN (Uy et al., 2019) is a challenging dataset consisting of 15K 3D real objects across 15 categories from the indoor scene dataset ScanNetv2 (Dai et al., 2017). Following the baselines, we use all three variants of ScanObjectNN: OBJ-BG, OBJ-ONLY, and PB-T50-RS. For all experiments, we sample 2048 points from each 3D shape and apply rotation augmentation. For the standard vanilla transformer, we fine-tune the model for 300 epochs with a batch size of 32 and a weight decay of 0.05. The learning rate is set to 2e-5 for OBJ-BG and OBJ-ONLY, and 5e-4 for the PB-T50-RS variant. For the Point-M2AE-based 3D encoder, we maintain the same settings, except that we use a learning rate of 5e-4 for OBJ-BG, 2e-5 for OBJ-ONLY, and 5e-4 for PB-T50-RS.

**ShapeNetPart.** ShapeNetPart (Yi et al., 2016) dataset consists of approximately 17K 3D objects from 16 categories with point-level annotations across 50 categories. We mostly follow the fine-tuning settings from Point-M2AE (Zhang et al., 2022a). We sample 2048 points from each shape and fine-tune the model for 300 epochs with a batch size of 16. The learning rate is set to 2e-4, and the weight decay is set to 0.1.

**S3DIS.** S3DIS (Armeni et al., 2016) dataset consists of six large-scale indoor areas containing a total of 273 million points from 13 categories, and it provides dense annotations for the segmentation task. Following ACT (Dong et al., 2022), we select Area 5 for evaluation. We fine-tune the model for 60 epochs with a batch size of 32. The AdamW optimizer is used with a weight decay of 1e-5, and a cosine learning rate scheduler is applied with a learning rate of 2e-5.

**ScanNetv2** We follow the same training recipe and architecture as 3DTER (Misra et al., 2021) and 3DTER-m (Zhang et al., 2022a) for fine-tuning on ScanNetv2 for 3D object detection.

## B Additional Results on Part Segmentation and Detection

### B.1 Results on Part Segmentation

We evaluate the performance of our pre-trained approach in Part Segmentation using ShapeNetPart (Yi et al., 2016). This settings requires the model to understand local patterns by predicting part labels for each point. We follow PointM2AE (Zhang et al., 2022a) for segmentation head and other experimental details. We compare the performance of our approach with other baselines in Table 9 and report both the mean IoU over all categories and the mean IoU over all instances. Our approach achieves the SOTA performance and improves Point-M2AE by $+0.6\%$ and $+0.7\%$ on both metrics respectively.

### B.2 Results on 3D Object Detection

We further conduct experiments on 3D object detection using ScanNetv2 (Dai et al., 2017) dataset. Following ACT (Dong et al., 2022), we use 3DETR (Misra et al., 2021), which has a transformer encoder consisting of three blocks and a transformer decoder. We compare various methods using mean Average Precision (mAP) at two different IoU thresholds of 0.50 and 0.25 with the results reported in Table 10. Our approach improves the 3DETR baseline by $+5.5\%$ and $+2.4\%$ on $AP_{50}$ and $AP_{25}$, respectively. Compared to other SSL methods including ACT, our approach outperforms them by a significant margin. In addition to conducting experiments with 3DETR, we also evaluate the approach using 3DTER-m, following Point-M2AE. We keep the implementation and architecture details consistent with those of Point-M2AE. Our two-stage pre-training approach boost the performance of 3DETR-m by $+2.4\%$ on $AP_{50}$ and $+2.5\%$ on $AP_{25}$, clearly outperforming Point-M2AE, as reported in Table 10.

## C Additional Ablation Study

**Predicting only masked tokens.** In Table 12, we conduct an ablation study to investigate the effect of performing local distillation only on masked tokens. The results show that, for both ModelNet40 and the OBJ_BG variant of ScanObjectNN, performing local distillation solely on masked tokens marginally degrades performance. Therefore, we apply local distillation to all tokens, along with token shuffling scheme.

Table 9: **Part Segmentation Results on ShapeNetPart dataset.** We report mean IoU across all categories (Cls.mIoU) and mean IoU across all instances (Inst.mIoU). † means the transformer is pre-trained with ImageNet dataset.

| Methods | Part Seg. | |
|---|---|---|
| | Cls.mIoU | Inst.mIoU |
| PointNet Qi et al. (2017a) | 80.4 | 83.7 |
| PointNet++ Qi et al. (2017b) | 81.9 | 85.1 |
| DGCNN Wang et al. (2019) | 82.3 | 85.2 |
| PointMLP Ma et al. (2022) | 84.6 | 86.1 |
| *with Single-Modal Self-Supervised Representation Learning* | | |
| Transformer Vaswani et al. (2017) | 83.4 | 84.7 |
| CrossPoint Afham et al. (2022) | - | 85.5 |
| Point-BERT Yu et al. (2022) | 84.1 | 85.6 |
| MaskPoint Liu et al. (2022b) | 84.4 | 86.0 |
| Point-MAE Pang et al. (2022) | 84.2 | 86.1 |
| Point-M2AE Zhang et al. (2022a) | 84.9 | 86.5 |
| PointGPT Chen et al. (2024) | 84.1 | 86.2 |
| Point-FEMAE Zha et al. (2023a) | 84.9 | 86.3 |
| PViT Qian et al. (2024) | 83.7 | 85.7 |
| *with Cross-Modal Self-Supervised Representation Learning* | | |
| ACT Dong et al. (2022) | 84.7 | 86.1 |
| ReCon Qi et al. (2023) | 84.8 | 86.4 |
| **Ours (Point-M2AE)** | **85.5** | **87.2** |
| *Improve (over Point-M2AE)* | +0.6 | +0.7 |
| Mamba3D Han et al. (2024) | 83.6 | 85.6 |
| PViT+Pix4Point Qian et al. (2024)† | **85.6** | 86.8 |

Table 10: **3D object detection on the ScanNetV2 dataset.** We report Average Precision (AP) at two different IoU thresholds of 0.50 and 0.25.

| Method | SSL | Input | $AP_{50}$ | $AP_{25}$ |
|---|---|---|---|---|
| VoteNet (Qi et al., 2019) | × | *xyz* | 33.5 | 58.6 |
| PointContrast (Xie et al., 2020) | ✓ | *xyz* | 38.0 | 59.2 |
| STRL (Huang et al., 2021) | ✓ | *xyz* | 38.4 | 59.5 |
| RandomRooms (Rao et al., 2021) | ✓ | *xyz* | 36.2 | 61.3 |
| DepthContrast (Zhang et al., 2021) | ✓ | *xyz* | - | 61.3 |
| 3DETR (Misra et al., 2021) | × | *xyz* | 37.9 | 62.1 |
| Point-BERT (Yu et al., 2022) | ✓ | *xyz* | 38.3 | 61.0 |
| MaskPoint (Liu et al., 2022b) | ✓ | *xyz* | 40.6 | 63.4 |
| ACT Dong et al. (2022) | ✓ | *xyz* | 42.1 | 63.8 |
| Multi-View ML Chen et al. (2025) | ✓ | *xyz* | 43.3 | 63.9 |
| **Ours (3DETR)** | ✓ | *xyz* | **43.4** | **64.5** |
| 3DTER-m Misra et al. (2021) | × | *xyz* | 47.0 | 65.0 |
| Point-M2AE Zhang et al. (2022a) | ✓ | *xyz* | 48.3 | 66.3 |
| **Ours (3DETR-m)** | ✓ | *xyz* | **49.4** | **67.5** |

**Masking Ratio.** In Figure 2, we present the ablation study of different masking ratio using both vanilla transformer and multi-scale hierarchical architecture. We report the results on PB-T50-RS variant of the

Table 11: **Effect of different teachers.** Overall accuracy (%) without voting is reported.

| Teacher(s) | MN40 | ScanOBJNN |
|---|---|---|
| CLIP only | 93.8 | 88.9 |
| DINOv2 only | 93.5 | 88.2 |
| CLIP + DINOv2 (ours) | **94.0** | **90.1** |

Table 12: **Effect of applying loss only on masked tokens in the second stage pre-training.** Overall accuracy (%) without voting is reported.

| Prediction Tokens | Model | MN40 | OBJ_BG |
|---|---|---|---|
| Masked tokens | Point-M2AE | 94.2 | 95.9 |
| All | Point-M2AE | **94.3** | **96.1** |

ScanObjectNN dataset. It can be observed that the masking ratios yielding the best performance are consistent with those found in both Point-MAE (Pang et al., 2022) and Point-M2AE (Zhang et al., 2022a).

**Different loss function for distillation.** Table 13 shows the ablation study on distillation objective. We explore various loss functions to select the most suitable one for the second stage of pre-training and find that the cosine similarity yields the best performance on both ModelNet40 and ScanObjectNN.

**Decoder Depth.** In Table 14, 15, we explore the different decoder networks and the different decoder depths for both local and global distillation. The left table of the Table 14 presents the results on ModelNet40 using the standard transformer architecture for local distillation. We find that performance is not sensitive to the decoder depth and the decoder with 4 blocks achieves the best performance. For global distillation objective, the decoder with 2 blocks achieves the highest results as seen in the right table of the Table 14. We also perform the same ablation study for multi-scale hierarchical architecture based pre-training in Table 15. For local distillation, the decoder with 2 stages yields the better results which is consistent with Point-M2AE as shown in left table of Table 15. For global distillation, we find that a single linear layer yields the best performance, which is denoted as zero decoder depth in the right table of Table 15.

**Choice of teacher models.** In Figure 3, we show the zero-shot accuracy (%) on ModelNet40 using different variants of CLIP and DINOv2. We use Point-M2AE based multi-scale hierarchical model as 3D encoder. It can be seen that combination of larger teachers yield the best performance. More specifically, combination of base models of CLIP and DINOv2 achieve 70.9 top-1 accuracy which outperforms ULIP-2 (Xue et al., 2024)

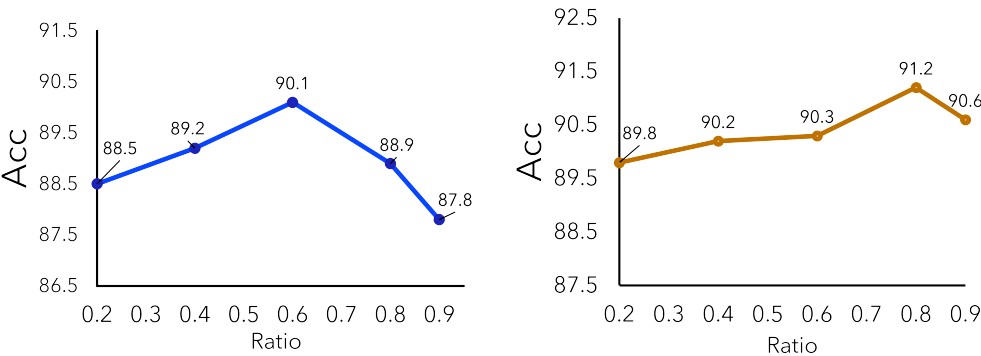

Figure 2: **Ablation study on the masking ratio for Point-MAE and Point-M2AE during second stage of the pre-training.** Overall accuracy (%) without voting is reported.

Table 13: **Ablation study of different loss functions used for feature distillation.** Overall accuracy (%) without voting is reported.

| Loss Function | ModelNet40 | OBJ_BG |
|---|---|---|
| MSE Loss | 94.00 | 95.35 |
| Smooth $\ell_1$ | 94.15 | 95.68 |
| Cosine Similarity | **94.30** | **96.10** |

Table 14: **Ablation study on the depth of the decoders used in Point-MAE for local and global distillation objective, respectively.** Overall accuracy (%) without voting is reported.

| Dec. Depth | Acc (%) |
|---|---|
| 1 | 93.87 |
| 2 | 93.91 |
| 4 | **94.00** |

| Dec. Depth | Acc (%) |
|---|---|
| 1 | 93.93 |
| 2 | **94.00** |
| 4 | 93.89 |

that achieves 69.7 top-1 accuracy using SLIP (Mu et al., 2021) ViT-B as reported in the paper. Table 11 compares the effect of using different teachers. CLIP alone achieves higher accuracy than DINOv2, likely because its joint image–text pretraining provides richer semantic cues. DINOv2, while strong on dense spatial features, relies only on image supervision and performs slightly worse in isolation. Combining CLIP and DINOv2 yields the best results, confirming that global semantics and dense spatial features are complementary and lead to stronger downstream performance.

**Choice of Pooling Operation.** In Table 16, we investigate different pooling operations to integrate features of all point tokens. We pick Point-M2AE based 3D encoder and find that the *max + average pooling* performs the best during fine-tuning which is consistent with Point-M2AE. We also use the same configuration during stage-1 pre-training and observe that it performs the best under zero-shot setting on ModelNet40.

# D    Comparison with ReCon

Our approach introduces key distinctions. Unlike ReCon, which primarily relies on a single vision-language model, our framework leverages multiple foundation models (CLIP and DINOv2) to enhance both global semantic and dense spatial representations. Additionally, our two-stage design uniquely integrates global-local feature distillation, ensuring better retention of multimodal semantics while refining 3D geometric features

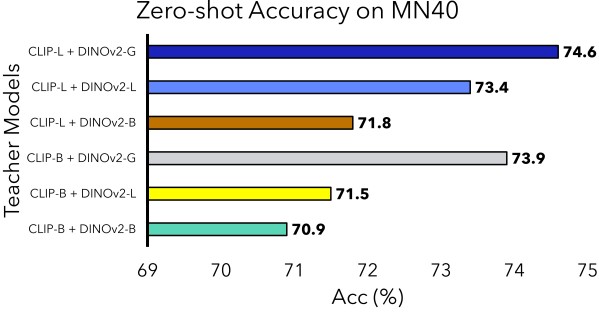

Figure 3: **Ablation study on the different combination of the teachers used in the first stage.** Zero-shot accuracy (%) on ModelNet40 is reported.

Table 15: **Ablation study on the depth of the decoders used with multi-scale hierachical 3D encoder for local and global distillation objective.** Overall accuracy (%) without voting is reported.

| Dec. Stages | Acc (%) |
|:---:|:---:|
| 1 | 93.6 |
| 2 | **94.3** |
| 3 | 92.5 |

| Dec. Depth | Acc (%) |
|:---:|:---:|
| 0 | **94.30** |
| 1 | 94.21 |
| 2 | 94.16 |

Table 16: **Different pooling operation used in Point-M2AE.** We report overall accuracy (%) without voting on ModelNet40 (MN40) under zero-shot setting and ScanObjNN-OBJ_BG. ZS stands for zero-shot setting.

| Pooling Op. | ZS-MN40 | MN40 | OBJ_BG |
|:---:|:---:|:---:|:---:|
| Max Pool | 72.7 | 93.7 | 95.1 |
| Average Pool | 72.3 | 93.4 | 94.8 |
| MAP Lee et al. (2019) | 73.3 | 94.0 | 95.8 |
| Max + Avg. Pool | **73.4** | **94.3** | **96.1** |

Table 17: We compare the fine-tuning costs with ReCon using NVIDIA A100.

| Method | #Params | Fine-tune GPU hours | GFLOPS | ScanObjectNN-OBJ_BG |
|:---:|:---:|:---:|:---:|:---:|
| ReCon | 43.6M | 2.3h | 5.3 | 95.18 |
| Ours (Point-MAE) | 22.1M | 2h | 4.8 | 95.20 |
| Ours (Point-M2AE) | 12.9M | 1.95h | 3.6 | **96.10** |

which we have demonstrated empirically on the benchmark datasets. In contrast, ReCon employs a generative student to guide a contrastive student within a single-stage pipeline.

## E   t-SNE visualization

The figures 4 and 5 show the t-SNE (Van der Maaten & Hinton, 2008; Poličar et al., 2019) feature manifold visualization of models before and after fine-tuning on the ModelNet40 and ScanObjectNN datasets. It can be observed that after fine-tuning, the features become more discriminative for both datasets. Another observation from the figures is that the pre-trained model can also produce discriminative features on ModelNet40, likely due to the smaller domain gap between ShapeNet and ModelNet40. In contrast, the domain gap is larger between ShapeNet and ScanObjectNN, which results in less discriminative features for the pre-trained model.

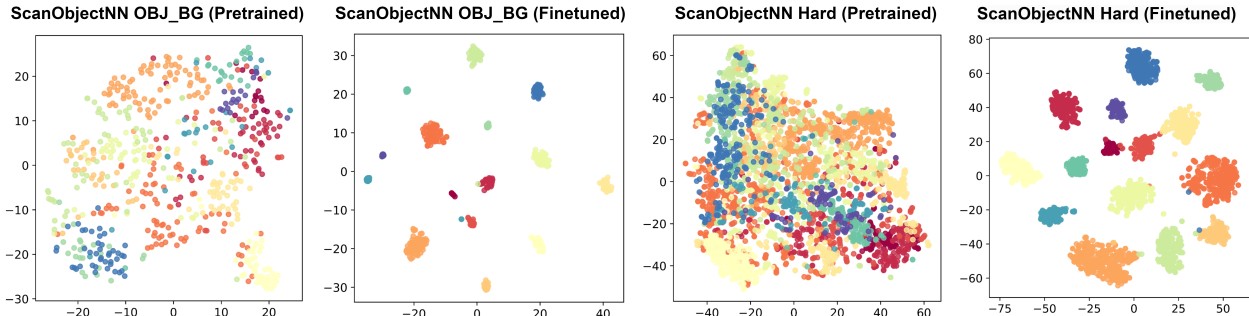

Figure 4: t-SNE (Van der Maaten & Hinton, 2008) visualization on ScanObjectNN variants using feature extracted from pre-trained model on ShapeNet and fine-tuned model on ScanObjectNN variants. We use multi-scale hierarchical transformer architecture.

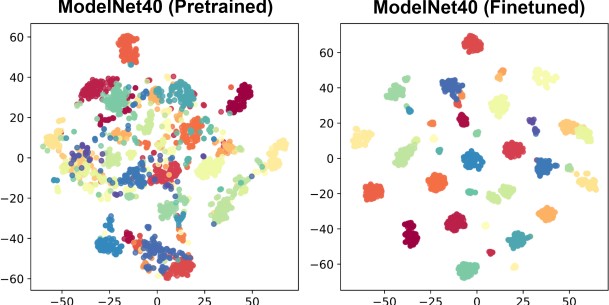

Figure 5: t-SNE (Van der Maaten & Hinton, 2008) visualization on ModelNet40 using feature extracted from pre-trained model on ShapeNet and fine-tuned model on ModelNet40. We use standard transformer architecture.