# OpenReview forum: "Multimodal Masked Point Distillation for 3D Representation Learning"
_TMLR — Accepted by TMLR_

### Review · Reviewer_Vsoq · 2025-12-30

**Summary Of Contributions:**

This paper proposes a novel two-stage pre-training framework for 3D representation learning that aligns point cloud features with multiple vision and language foundation models before performing enhanced masked point modeling. The approach introduces a global-local feature distillation strategy and a token shuffling mechanism in the second stage to effectively transfer multimodal semantic knowledge while capturing intricate 3D geometric relationships. Experimental results demonstrate that this model-agnostic method achieves state-of-the-art performance across various benchmarks, including 3D object recognition, semantic segmentation, and object detection.

**Audience:**

Yes

**Audience Explanation:**

The proposed method provides a scalable and model-agnostic framework for enhancing 3D point cloud understanding through multimodal alignment and a novel two-stage distillation process. These findings will be of great interest to the TMLR audience as they offer significant advancements in cross-modal representation learning and demonstrate state-of-the-art performance across various standard 3D benchmarks.

**Broader Impact Concerns:**

I do not see any specific ethical concerns or negative societal impacts arising from this work.

**Claims And Evidence:**

Yes

**Claims Explanation:**

The authors provide comprehensive empirical evidence through extensive evaluations across multiple standard 3D benchmarks, consistently achieving state-of-the-art performance in tasks ranging from object recognition to semantic segmentation. Furthermore, the detailed ablation studies effectively validate the individual contributions of the proposed two-stage framework, demonstrating that the global-local distillation and token shuffling mechanisms are essential for the observed performance gains.

**Requested Changes:**

1. The proposed two-stage pre-training process significantly increases the total training time, taking approximately 3.5 times longer than the baseline Point-MAE.

2. The model relies heavily on ShapeNet for pre-training, which is a synthetic dataset that may not fully represent the noise and complexity found in real-world sensor data. Testing the framework with larger, more diverse real-world pre-training datasets like Objaverse-XL could better demonstrate its scalability and robustness.

3. While the paper advocates for using multiple vision foundation models, it lacks a detailed ablation study comparing the performance gain of using an ensemble versus a single strong teacher like DINOv2. This makes it difficult to determine if the increased complexity of managing multiple teachers is truly necessary for the observed performance.

4. The current experiments are confined to object-level recognition and indoor semantic segmentation, leaving the model's performance on large-scale outdoor LiDAR data (e.g., KITTI or Waymo) unknown. Evaluating the model on outdoor autonomous driving benchmarks would be crucial to confirm its broader applicability across all 3D domains.

5. The reliance on BLIP-2 and fixed prompt templates might still restrict the semantic variety and linguistic complexity of the features learned in Stage 1. Incorporating a wider range of human-generated or more descriptive captions could potentially improve the model's zero-shot capabilities and semantic understanding.

6. By focusing on feature distillation rather than direct coordinate reconstruction, the model may prioritize high-level semantics over capturing precise local geometric structures. A comparative analysis of reconstruction quality versus distillation accuracy would help clarify whether important structural information is being sacrificed.

7. Although the framework is claimed to be model-agnostic, the experimental validation is primarily focused on Transformer-based architectures like Point-M2AE. Testing the approach on other common 3D backbones, such as Sparse CNNs or MLP-based models, would provide stronger evidence for its universal utility.

8. The paper does not offer an extensive analysis of how sensitive the results are to the weights of the local and global distillation losses or the distillation temperature.

9. Stage 1 uses rendered 2D images from a fixed set of viewpoints, which might result in missing structural information from occluded regions of the 3D objects. Implementing a more dynamic or multi-view rendering strategy could ensure a more holistic alignment between the 2D modalities and the 3D point cloud.

---

> ### Author Response · Authors · 2026-01-29
> **Response to Reviewer Vsoq**
>
> We thank the reviewer for the thorough and constructive feedback. Below we address each concern and clarify the scope, motivation, and additional experimental evidence.
>
> $\textbf{1 - Increased training cost of the two-stage framework}$
>
> We acknowledge the extra pretraining time (3.5× over Point-MAE), but this is a one-time cost, as inference and fine-tuning FLOPs/parameters remain comparable. The added cost is justified by consistent gains across 3D tasks, including +4–5\% on ScanObjectNN and SOTA zero-shot/transfer results. Notably, simply extending Point-MAE training by 3.5× does not achieve similar accuracy (Table 5). The framework is model-agnostic, scalable, and introduces no custom layers.
>
> $\textbf{ 2- Reliance on ShapeNet and synthetic data}$
>
> We agree that large-scale real-world datasets such as Objaverse-XL are valuable for pretraining, as already discussed in Section 6. In this work, our experiments focus on standard-sized backbones and medium-scale datasets due to computational constraints, allowing us to isolate the effect of the proposed training framework. Despite being synthetic, ShapeNet is a widely used benchmark for 3D representation learning, and our pretrained models generalize well to real-world datasets such as ScanObjectNN, S3DIS, and ScanNetv2. We leave large-scale real-world pretraining as an important direction for future work.
>
> $\textbf{3 - Necessity of multi-teacher supervision}$
>
> We already include ablation studies comparing CLIP-only, DINOv2-only, and CLIP + DINOv2 supervision (Table 11 and Figure 3 in the supplementary material), which show that neither teacher alone matches the performance of their combination. We will further clarify in the revised manuscript that the motivation for multi-teacher alignment is not ensembling at inference time, but rather injecting complementary semantic (CLIP) and dense spatial (DINOv2) cues into the 3D encoder during pretraining.
>
>
> $\textbf{4 - Lack of outdoor LiDAR evaluation}$
>
> We agree that outdoor LiDAR benchmarks are an important application domain. Our current focus is on object-centric and indoor scene understanding tasks, which align with the datasets and multimodal supervision used in Stage-1. Extending the framework to large-scale outdoor LiDAR data would require domain-specific considerations (e.g., long-range sparsity, temporal aggregation) and is beyond the scope of the current submission. We will clarify the intended application scope and highlight outdoor LiDAR as an important direction for future work.
>
> $\textbf{5 - Limited linguistic diversity due to BLIP-2 and fixed prompts}$
>
> Our use of BLIP-2 with fixed prompts follows prior work (e.g., ULIP-2) and provides a controlled way to study multimodal alignment. We agree that incorporating more diverse human-generated or long-form captions could further enrich semantic representations and potentially improve zero-shot performance. We will discuss this as a promising extension in the revised manuscript.
>
> $\textbf{ 6 - Feature distillation vs direct reconstruction}$
>
> We conduct an ablation comparing feature distillation and point-space reconstruction (cf. Table 8). The results show that feature distillation consistently outperforms coordinate reconstruction on ModelNet40 for both Point-MAE and Point-M2AE. Moreover, our empirical results on geometry-sensitive downstream tasks such as part segmentation, semantic segmentation, and 3D object detection, indicate that fine-grained local geometric information is preserved despite not directly reconstructing point coordinates.
>
> $\textbf{ 7 - Model-agnostic claim beyond Transformers}$
>
> We thank the reviewer for this point. While our initial experiments focused on Transformer-based architectures, we have additionally evaluated our framework on PointMamba, an SSM-based 3D encoder, and observe consistent improvements over the baseline. We will include these results in the revised manuscript and refine the scope of our model-agnostic claim accordingly.
>
> | Model                 |    OBJ_BG |  OBJ_ONLY | PB_T50_RS |
> | --------------------- | --------: | --------: | --------: |
> | Point-MAE             |     90.02 |     88.29 |     85.18 |
> | Ours (Point-MAE)  | 95.20 | 93.73 | 90.10 |
> | Point-M2AE            |     91.22 |     88.81 |     86.43 |
> | Ours (Point-M2AE) | 96.10 | 94.25 | 91.20 |
> | PointMamba            |     94.32 |     92.60 |     89.31 |
> | Ours (PointMamba) | 95.15 | 93.19 | 90.11 |
>
> $\textbf{8 - Sensitivity to distillation loss weights and temperature}$
>
> In our experiments, we found the framework to be relatively stable across a reasonable range of distillation loss weights. As a result, we fixed these weights to 1.0 during experimentation to reduce hyperparameter complexity. For the temperature in equation 1 and 2, we follow the ULIP2 to make it learnable. We will clarify this design choice and include a brief discussion of sensitivity trends in the revised manuscript.

---

> > ### Author Response · Authors · 2026-01-29
> > **Response to Reviewer Vsoq (Continue)**
> >
> > $\textbf{9 - Fixed multi-view rendering strategy in Stage-1}$
> >
> > We acknowledge this limitation and discuss it in Section 6. In this work, we adopt a fixed multi-view rendering strategy for consistency with prior work and to ensure controlled multimodal supervision. More dynamic or adaptive view sampling could further improve coverage of occluded regions and is an interesting direction for future work.

---

### Review · Reviewer_gNEr · 2026-01-11

**Summary Of Contributions:**

This paper proposes a model-agnostic, two-stage pre-training framework for point cloud representation learning that effectively integrates semantic knowledge with geometric understanding. In the first stage, the authors employ multiple pre-trained vision and language foundation models as teachers to align 3D encoder features with multimodal representations via contrastive learning. The second stage refines these representations through an improved masked point modeling approach, which incorporates a novel token shuffling strategy and global-local feature distillation to capture local geometry while preventing shortcut learning and preserving semantic information. Extensive experiments demonstrate that this method achieves state-of-the-art performance on various benchmarks, notably reaching 96.1% accuracy on ScanObjectNN (OBJ-BG) when using a Point-M2AE backbone.

**Audience:**

Yes

**Audience Explanation:**

The proposed two-stage framework, which integrates multimodal alignment with improved masked point modeling, offers a practical solution for enhancing 3D encoders. The strong empirical results on standard benchmarks (e.g., ScanObjectNN) and the detailed ablation studies on distillation strategies (specifically token shuffling and global-local distillation) would be of significant interest to researchers working on 3D computer vision, self-supervised learning, and multimodal foundation models.

**Broader Impact Concerns:**

None. The paper focuses on technical improvements in 3D representation learning and does not raise specific ethical concerns.

**Claims And Evidence:**

Yes

**Claims Explanation:**

The method is model-agnostic and achieves state-of-the-art results on several benchmarks, including a notable 96.1% accuracy on ScanObjectNN (OBJ-BG) when using Point-M2AE.

**Requested Changes:**

The first stage relies on multimodal contrastive learning. This paradigm is already quite mature in the 3D domain (e.g., ULIP, OpenShape). While the authors use "multiple teachers", the paper does not provide sufficient insight or theoretical analysis on why this specific multi-teacher setup provides a fundamental distinction or advantage over existing single-teacher cross-modal methods. The innovation here feels somewhat incremental.

A key trend in current representation learning is validating the "scaling laws" (e.g., Uni3D scaling to 1B parameters). This work largely operates on standard-sized backbones like Point-MAE and Point-M2AE. The paper does not explore how this two-stage approach scales with larger model sizes or massive datasets, which limits the evaluation of its potential as a true "foundation" model.

he paper claims the approach is "model-agnostic" and applicable to "any 3D transformer encoder". However, with the recent rise of State Space Models (SSMs) in 3D vision (e.g., PointMamba), it is important to know if this mask-and-distill strategy generalizes beyond Transformers. Validating this on Mamba-based architectures would significantly strengthen the claim of universality.

The comparison in Table 3  appears to miss some very recent state-of-the-art works. For instance, PointGST (published in IEEE TPAMI) has reported extremely high performance on ScanObjectNN (OBJ-BG), reportedly reaching around ~99%. Omitting such strong, recent baselines weakens the claim of achieving state-of-the-art results.

---

> ### Author Response · Authors · 2026-01-29
> **Response to Reviewer gNEr**
>
> We thank the reviewer for the constructive feedback and we address each concern below with clarifications and additional experimental evidence.
>
> $\textbf{1 - On the novelty of multi-teacher multimodal alignment}$
>
> We agree that multimodal contrastive learning itself is a mature paradigm in 3D representation learning (e.g., ULIP, ULIP2, OpenShape). Our contribution is not to introduce a new contrastive objective, but to show that the choice and combination of teachers materially changes the nature of the learned 3D representation and enables the subsequent geometry-focused distillation stage.
> Specifically, prior methods rely on a single teacher (typically CLIP), which emphasizes global semantic alignment. We show empirically that combining CLIP (global semantics) with DINOv2 (dense spatial features) yields complementary supervision that neither teacher provides alone. This is validated through ablations (see Table 11 and Figure 3 in appendix), where each teacher individually underperforms their combination across zero-shot and fine-tuning benchmarks.
> Importantly, the role of multi-teacher alignment in our framework is instrumental rather than standalone: it creates a 3D encoder whose representations are sufficiently rich to serve as a semantic teacher for Stage-2 masked distillation. Single-teacher alignment does not yield the same gains in the second stage. We will clarify this motivation and explicitly position multi-teacher alignment as a means to enable effective cross-stage distillation, rather than as an isolated contribution.
>
> $\textbf{2 - On scaling laws and foundation model claims}$
>
> We agree that studying scaling laws with billion-parameter models and massive datasets is an important direction. Our work does not aim to compete directly with large-scale efforts such as Uni3D, which focus on architectural scaling and dataset expansion. We note that this limitation is already explicitly discussed in the paper (Discussion and Limitations) while our experiments are focused on standard-sized backbones due to computational constraints, and that scaling to larger models and datasets is left for future work. Instead, our goal is to propose a training framework that is compatible with existing and future architectures, and to demonstrate its effectiveness across multiple standard backbones and tasks under realistic compute budgets. We therefore focus on widely adopted architectures (Point-MAE, Point-M2AE, 3DETR-based encoders) to isolate the effect of the training strategy itself.
>
> $\textbf{3 - On generalization beyond Transformers (e.g., Mamba / SSMs)}$
>
> We thank the reviewer for raising this point. In response, we have conducted additional experiments by applying our two-stage pretraining framework to a PointMamba, with minimal hyperparameter tuning and architectural design. We observe consistent performance improvements over the corresponding baseline, indicating that our framework can generalize beyond transformer-based encoders and is compatible with SSM-style backbones.
>
> We will include these results in the revised manuscript and clarify that, while further optimization and large-scale validation are promising directions, these initial findings already demonstrate that our approach is model agnostic.
>
> | Model                 |    OBJ_BG |  OBJ_ONLY | PB_T50_RS |
> | --------------------- | --------: | --------: | --------: |
> | Point-MAE             |     90.02 |     88.29 |     85.18 |
> | Ours (Point-MAE)  | 95.20 | 93.73 | 90.10 |
> | Point-M2AE            |     91.22 |     88.81 |     86.43 |
> | Ours (Point-M2AE) | 96.10 | 94.25 | 91.20 |
> | PointMamba            |     94.32 |     92.60 |     89.31 |
> | Ours (PointMamba) | 95.15 | 93.19 | 90.11 |

---

> > ### Comment · Reviewer_gNEr · 2026-01-29
> >
> > Thanks. My major concerns have been addressed well.

---

> ### Author Response · Authors · 2026-01-29
> **Response to Reviewer gNEr (Continue)**
>
> $\textbf{4 - On missing recent baselines (e.g., PointGST)}$
>
> We thank the reviewer for pointing out PointGST, a recent work that reports very high performance on ScanObjectNN. PointGST focuses on parameter-efficient fine-tuning via spectral-domain adapters applied to a frozen pretrained backbone. They report high performance on ScanNet using PointGPT-L as a pre-trained model. In response, we have also evaluated PointGST using our pretrained PointMAE as the backbone and applied its spectral-domain fine-tuning strategy, and we report the resulting performance in the following table. Table shows that our pretrained PointMAE remains compatible with PointGST fine-tuning, yielding additional gains. These results will be included in the revised manuscript.
>
> | Model                      | OBJ_BG | OBJ_ONLY | PB_T50_RS |
> | -------------------------- | -----: | -------: | --------: |
> | Point-MAE                  |  90.02 |    88.29 |     85.18 |
> | Point-M2AE                 |  91.22 |    88.81 |     86.43 |
> | ACT                        |  93.29 |    91.91 |     88.21 |
> | ReCon                      |  95.18 |    93.63 |     90.63 |
> | ReCon + PointGST           |  94.49 |    92.94 |     89.49 |
> | Ours (PointM2AE)           |  96.10 |    94.25 |     91.20 |
> | Ours (PointMAE)            |  95.20 |    93.73 |     90.10 |
> | Ours (PointMAE + PointGST) |  95.36 |    94.11 |     90.18 |
> | PointGPT-L + PointGST      |  **98.97** |    **97.59** |     **94.83** |

---

### Review · Reviewer_zxLL · 2026-01-15

**Summary Of Contributions:**

This paper proposes a two-stage pre-training stage taking point clouds as input for some 3D understanding tasks.

The first stage pre-trains the 3D encoder by aligning the 3D representation with representations of other modalities, like vision and language extracted by several pre-training foundation models.

The second stage conducts masked point modeling using global-local feature distillation of semantic 3D embeddings together with token shuffling.

SOTA performance is achieved on some common benchmarks (ModelNet40, ScanObjectNN, S3DIS, ScanNetv2, ShapeNetPart) across 3D recognition tasks (point cloud classification, segmentation, and detection) over previous methods (like Point-MAE and Point-M2AE).

**Audience:**

Yes

**Audience Explanation:**

I think people who are working on point cloud representation learning and benefit from this paper, including the proposed method and experimental results.

**Broader Impact Concerns:**

There are no significant ethical concerns in this paper.

**Claims And Evidence:**

Yes

**Claims Explanation:**

This paper is generally well-written. Experiments are extensive and thorough. The proposed idea is not fundamentally new and novel, but it seems reasonable and achieves reasonable performance over baseline and previous methods. Most claims are well-justified.

**Requested Changes:**

As I mentioned above, this paper positions itself against ACT, ReCon, ReCon++, and related two-stage or teacher–student frameworks, the actual conceptual novelty remains not clear. For example:

1) There is no significant distinction between the Stage-2 “local + global feature distillation” and other feature-space distillation methods (like ACT and I2P-MAE)
2) It’s not so clear which part of the proposed method is fundamentally new (like token shuffling as an anti-shortcut, or multimodal alignment using many foundational models) versus engineering of known ideas.

A subsection that clearly explains what is new compared with other methods could be helpful.

---

> ### Author Response · Authors · 2026-01-24
> **Response to Reviewer zxLL**
>
> We thank the reviewer for the thoughtful comments and for highlighting the need to clarify the conceptual novelty of our approach relative to other frameworks (e.g., ULIP-2, ACT, ReCon).
>
>
> While our framework builds upon prior directions (ULIP-2, ACT, ReCon), our contribution lies in the specific integration and problem decomposition of these ideas into a unified two-stage pretraining pipeline. Our main novelty is a two-stage framework that combines multi-teacher multimodal alignment with geometry-focused masked point distillation for 3D point clouds, which has not been explored in prior work. Unlike ULIP-2, which relies on a single CLIP teacher, we demonstrate that combining CLIP (global semantic alignment) and DINOv2 (dense spatial cues) yields complementary supervision that consistently improves downstream performance. This constitutes a form of multi-teacher distillation for 3D pretraining, which has not been previously studied. Furthermore, while inspired by permutation-based pretext tasks (e.g., Jigsaw), we incorporate token shuffling within masked point modeling and cross-stage feature distillation, which is novel in the 3D setting and empirically improves robustness by preventing shortcut learning.
>
> Below, we further clarify the distinctions between our approach and ACT, I2P-MAE, and ReCon.
>
> $\textbf{Clarification relative to I2P-MAE.}$
>
> I2P-MAE transfers 2D knowledge by incorporating saliency-guided masking and 2D semantic reconstruction directly into the masked autoencoding objective. In this design, multimodal signals are tightly coupled to the decoder and reconstruction losses, and there is no standalone 3D semantic teacher. In contrast, our approach explicitly decouples multimodal knowledge acquisition from masked point modeling. We first learn a 3D encoder via multimodal alignment with multiple foundation models (Stage-1), and then distill its global and token-level representations during geometry-focused masked modeling (Stage-2). This separation preserves multimodal semantics while avoiding reconstruction-specific biases, and enables our framework to be applied to different MAE-based 3D encoders without redesigning masking or reconstruction objectives.
>
> $\textbf{Clarification relative to ACT.}$
>
>
> While ACT also employs feature-space supervision for masked point modeling, its design differs fundamentally from ours. In ACT, feature distillation is performed from a cross-modal autoencoding teacher, and supervision is applied primarily to masked tokens within a tightly coupled pipeline. In contrast, our teacher is a Stage-1 3D encoder trained purely via multimodal alignment using multiple foundation models, decoupled from reconstruction objectives. Moreover, we distill both global and token-level representations, and introduce token shuffling prior to decoding to explicitly prevent shortcut learning on visible tokens—an aspect not addressed in ACT. Our ablation studies show that these design choices are critical for performance.
>
> Another key distinction is architecture generality. While ACT allows changing the pretrained 2D transformer, the 3D teacher–student pipeline itself is tied to a specific 3D autoencoding architecture and integrates only a single pretrained model at a time, limiting extensibility. In contrast, our two-stage formulation treats the Stage-1 3D encoder as a standalone semantic teacher and applies masked distillation in Stage-2 without modifying the encoder design. As a result, our method is backbone-agnostic and can be directly applied to different MAE-based architectures, including Point-MAE, Point-M2AE, and 3DETR-based models, which we validate empirically across multiple backbones and tasks. Thus, our contribution is not a new backbone, but a general pretraining framework for 3D transformers.
>
> $\textbf{Comparison with ReCon.}$
>
>
> A detailed comparison with ReCon is provided in Section C of the supplementary material, and we will move a summarized version of this discussion to the main paper in the revised submission to further improve clarity.

---

### Review · Reviewer_SseK · 2026-01-26

**Summary Of Contributions:**

This manuscript introduces a two-stage self-supervised pre-training framework for 3D point cloud representation learning. The first stage aligns point cloud features with image, depth, and text embeddings using multiple vision and language foundation models, aiming to enrich semantic representations beyond single-teacher paradigms. The second stage focuses on point-only learning via enhanced masked point modeling, incorporating token shuffling and local/global feature distillation to preserve semantic knowledge while strengthening geometric reasoning.

The approach is model-agnostic and demonstrated on Point-MAE and Point-M2AE backbones. Extensive experiments across object classification, semantic segmentation, part segmentation, and 3D object detection show consistent improvements over strong baselines (e.g., Point-MAE, ACT, ReCon), with particularly notable gains on ScanObjectNN.

Overall, the work is well motivated and empirically solid, though its novelty is primarily incremental, arising from the integration and staging of existing techniques.

**Audience:**

Yes

**Audience Explanation:**

Yes. The manuscript will interest researchers in self-supervised learning, multimodal learning, and 3D vision, particularly those developing general-purpose 3D encoders. The model-agnostic design and broad evaluation make the findings practically relevant, even if the contribution is more incremental than foundational.

**Broader Impact Concerns:**

No major ethical concerns are identified. As the method relies on large pre-trained multimodal models, a brief acknowledgment of potential inherited biases and their downstream implications would be sufficient if a Broader Impact Statement is required.

**Claims And Evidence:**

Yes

**Claims Explanation:**

Yes. The claims are largely supported by comprehensive experimental results across multiple datasets and tasks. The reported improvements are consistent and substantiated through careful ablations that analyze the impact of key components such as token shuffling and feature distillation.

**Requested Changes:**

- **RC1:** Please clarify the novelty relative to prior two-stage and distillation-based methods (e.g., ACT, ReCon), as well as those from LiDAR-based 3D representation learning tasks (e.g., SLidR, Seal, SuperFlow, LargeAD).

- **RC2:** Please add discussion or analysis on the robustness and quality of multimodal supervision, especially BLIP-2–generated captions and multi-teacher interactions.

Would strengthen the work for:
- **RC3:** (Minor) Include additional qualitative visualizations to illustrate semantic transfer.

- **RC4:** (Minor) Discuss performance on more challenging or large-scale scene-level settings.

- **RC5:** (Minor) Improve figure and table readability.

---

> ### Author Response · Authors · 2026-02-02
> **Response to Reviewer SseK**
>
> We thank the reviewer for the constructive summary and suggestions and we address each concern below with clarifications.
>
> $\textbf{1 - Novelty relative to prior two-stage and distillation-based methods (ACT, ReCon) and LiDAR-based tasks (SLidR, Seal, SuperFlow, LargeAD)}$
>
> We will clarify the novelty of our approach by adding a dedicated comparison subsection in the main paper. While our framework builds upon prior directions, our contribution lies in the specific integration and problem decomposition of these ideas into a unified two-stage pretraining pipeline.
>
> Our main novelty is a two-stage framework that combines multi-teacher multimodal alignment with geometry-focused masked point distillation for 3D point clouds, which has not been explored in prior work. Unlike ULIP-2, which relies on a single CLIP teacher, we demonstrate that combining CLIP (global semantic alignment) and DINOv2 (dense spatial cues) yields complementary supervision that consistently improves downstream performance. This constitutes a form of multi-teacher distillation for 3D pretraining that has not been previously studied. Furthermore, while inspired by permutation-based pretext tasks (e.g., Jigsaw), we incorporate token shuffling within masked point modeling and cross-stage feature distillation, which is novel in the 3D setting and empirically improves robustness by preventing shortcut learning.
>
> Compared to ACT, although both approaches employ feature-space supervision, the designs differ fundamentally. In ACT, feature distillation is performed from a cross-modal autoencoding teacher and supervision is applied primarily to masked tokens within a tightly coupled pipeline. In contrast, our teacher is a Stage-1 3D encoder trained purely via multimodal alignment using multiple foundation models, decoupled from reconstruction objectives. Moreover, we distill both global and token-level representations, and introduce token shuffling prior to decoding to explicitly prevent shortcut learning on visible tokens—an aspect not addressed in ACT. Our ablation studies demonstrate that these design choices are critical for performance. Another key distinction is architecture generality. While ACT allows changing the pretrained 2D transformer, the 3D teacher–student pipeline itself is tied to a specific 3D autoencoding architecture and integrates only a single pretrained model at a time, limiting extensibility. In contrast, our two-stage formulation treats the Stage-1 3D encoder as a standalone semantic teacher and applies masked distillation in Stage-2 without modifying the encoder design. As a result, our method is backbone-agnostic and can be directly applied to different MAE-based architectures, including Point-MAE, Point-M2AE, and 3DETR-based models, which we validate empirically across multiple backbones and tasks. Thus, our contribution is not a new backbone, but a general pretraining framework for 3D transformers.
>
> A detailed comparison with ReCon is provided in Section C of the supplementary material, and we will move a summarized version of this discussion to the main paper in the revised submission to further improve clarity.
>
> We will also expand the comparison to LiDAR-based representation learning methods such as SLidR, Seal, SuperFlow, and LargeAD. These methods primarily rely on contrastive learning driven by spatial and temporal cues from sequential LiDAR data. In contrast, our framework employs multimodal contrastive alignment with multiple vision foundation models in Stage-1, and uses the resulting 3D encoder as a semantic teacher for Stage-2 masked point modeling via feature distillation. We will clarify that these approaches target different data regimes and supervisory signals, and discuss how they are complementary rather than directly comparable.
>
> $\textbf{2 - Robustness and quality of multimodal supervision (BLIP-2 captions and multi-teacher interactions)}$
>
> We will add a more detailed discussion on the robustness of multimodal supervision in Stage-1. Specifically, we will clarify the role of BLIP-2–generated captions as a scalable semantic signal, following prior work, and discuss their limitations. In addition, we compare Stage-1 supervision using manually constructed text prompts based on category names versus BLIP-2–generated descriptive captions in Table 2. As shown, BLIP-2–based supervision consistently outperforms manual category-level captions, indicating that richer and more descriptive language provides stronger semantic guidance. These findings suggest that incorporating even more diverse or human-generated captions could further improve performance, which we identify as a promising direction for future work.
>
> We will also expand the explanation of multi-teacher interactions, emphasizing that CLIP and DINOv2 provide complementary global semantic and dense spatial cues, respectively, and that the effectiveness of this design is supported by our ablation studies (Table 11 and Figure 3 in the supp. material).

---

> > ### Author Response · Authors · 2026-02-02
> > **Response to Reviewer SseK (Continue)**
> >
> > $\textbf{3 - Additional qualitative visualizations}$
> >
> > We agree that qualitative visualizations are helpful for understanding semantic transfer. We note that such analysis is already included in the supplementary material, where Figures 4 and 5 present t-SNE visualizations on ModelNet40 and ScanNet, comparing representations from pretrained and fine-tuned models. It can be observed that after fine-tuning, the features become more discriminative for both datasets. Another observation from the figures is that the pre-trained model can also produce discriminative features on ModelNet40, likely due to the smaller domain gap between ShapeNet and ModelNet40. In contrast, the domain gap is larger between ShapeNet and ScanObjectNN, which results in less discriminative features for the pre-trained model.
> >
> > $\textbf{4 - Discussion on larger or more challenging scene-level settings}$
> >
> > We will expand the discussion to more clearly articulate the current evaluation scope and limitations, and discuss how the proposed framework could extend to larger-scale or more challenging scene-level settings (e.g., outdoor or long-range scenes). As already discussed in Section 6 as a limitation of our approach, our experiments focus on standard-sized backbones and medium-scale datasets due to computational constraints, allowing us to isolate the effect of the proposed training framework. We will clarify that while such settings are beyond the scope of the current experiments, the framework itself is not inherently restricted to small-scale scenes.
> >
> > $\textbf{5- Improve figure and table readability}$
> >
> > Thanks for the suggestions. We will improve the readability of figures and tables in the revised manuscript.

---

### Decision · Action_Editor_Q9Pq · 2026-04-02

**Recommendation:** Accept with minor revision

**Audience:**

Yes

**Audience Explanation:**

This work addresses the important problem of learning general-purpose 3D representations by combining multimodal supervision with self-supervised geometric modeling, and is likely to be of interest to ML researchers working on foundation models, multimodal learning, and representation pre-training.

**Claims And Evidence:**

Yes

**Claims Explanation:**

This paper proposes a two-stage self-supervised pre-training framework for 3D point cloud representation learning. The method first aligns point cloud features with multimodal representations from multiple vision–language foundation models, and then refines them via enhanced masked point modeling with token shuffling and global–local feature distillation. Extensive experiments across multiple benchmarks demonstrate consistent performance gains over strong baselines.

All reviewers agree that the paper is well-motivated, technically sound, and empirically strong. The proposed framework is model-agnostic and shows consistent improvements across several 3D understanding tasks, supported by comprehensive ablations. Reviewers also highlighted the practical relevance of integrating multimodal supervision with geometric reasoning for general-purpose 3D representation learning.

The main concerns raised during the initial review phase focused on insufficient differentiation from prior two-stage or distillation-based approaches, unclear benefits of multi-teacher supervision, scalability and generalization claims, and missing comparisons with recent baselines. Some reviewers also noted additional issues such as training cost, reliance on synthetic pretraining data, and limited evaluation on non-transformer backbones or large-scale datasets.

During the discussion, the authors provided detailed clarifications on positioning relative to prior work (e.g., ACT, ReCon, I2P-MAE), expanded comparisons with recent methods, and added experimental evidence supporting generalization across architectures (including PointMamba). The rebuttal also addressed concerns regarding multimodal supervision, multi-teacher design, and robustness. Reviewers generally agreed that these clarifications and additional results adequately resolved their major concerns.

After the rebuttal, all reviewers expressed positive final assessments, with each reviewer leaning toward acceptance. The consensus is that, although the conceptual novelty is incremental, the method is carefully designed, empirically strong, and practically valuable for the 3D vision community. The additional experiments and clarifications strengthen the claims regarding generality and effectiveness.

Given the consistent empirical gains, solid technical quality, and satisfactory responses to reviewer concerns, I recommend acceptance. The authors should incorporate the additional discussion from the rebuttal into the final version.